# INJECTIVE FLOWS FOR STAR-LIKE MANIFOLDS

**Marcello Massimo Negri & Jonathan Aellen & Volker Roth**
Department of Mathematics and Computer Science, University of Basel

{marcellomassimo.negri, jonathan.aellen, volker.roth}@unibas.ch

## ABSTRACT

Normalizing Flows (NFs) are powerful and efficient models for density estimation. When modeling densities on manifolds, NFs can be generalized to injective flows but the Jacobian determinant becomes computationally prohibitive. Current approaches either consider bounds on the log-likelihood or rely on some approximations of the Jacobian determinant. In contrast, we propose injective flows for star-like manifolds and show that for such manifolds we can compute the Jacobian determinant exactly and efficiently. This aspect is particularly relevant for variational inference settings, where no samples are available and only some unnormalized target is known. Among many, we showcase the relevance of modeling densities on star-like manifolds in two settings. Firstly, we introduce a novel Objective Bayesian approach for penalized likelihood models by interpreting level-sets of the penalty as star-like manifolds. Secondly, we consider probabilistic mixing models and introduce a general method for variational inference by defining the posterior of mixture weights on the probability simplex.

## 1 INTRODUCTION

Normalizing Flows (NFs) are flexible and efficient models that allow us to accurately estimate arbitrary probability distributions. The key idea is to transform a simple base distribution into a complicated one through a series of bijections. However, in many applications we know that the target density lies on a certain lower-dimensional manifold. A common approach is to have the base distribution defined on the lower-dimensional space and use an injective transformation to embed it into higher dimensions. Unfortunately, the computation of the transformed density involves an expensive Jacobian determinant term, which makes the model computationally prohibitive. Currently, exact and efficient Jacobian determinant is possible only for trivial manifolds like spheres and tori (Gemici et al., 2016; Rezende et al., 2020) or for very restrictive transformations (Ross and Cresswell, 2021). In practice, most work renounce exact density estimation and approximate the Jacobian determinant term (Kumar et al., 2020; Kothari et al., 2021; Sorrenson et al., 2024), often with high variance estimators. Exact density estimation might not be critical when training with maximum likelihood, but is crucial in many applications like variational inference, where samples are not available.

In this paper, we introduce injective flows for star-like manifolds and show that we can exactly and efficiently compute the associated Jacobian determinant term, with the same computational cost as NFs. We consider star-like manifolds with intrinsic dimensionality $d-1$ and embedded in $\mathbb{R}^d$, which is a general class of manifolds that are particularly relevant in many statistical applications. Note that learning distributions, including the uniform distribution, on arbitrary manifolds is non-trivial and often requires complicated sampling schemes (Pennec, 2006; Diaconis et al., 2012). Distributions on the hypersphere are crucial in directional statistics, astrophysics, medicine, biology, meteorology and many other fields (Chikuse, 2012). Particularly relevant for geoscience applications is the fact that the Earth is an oblate spheroid, which is a star-like manifold. The probabilistic simplex, another star-like manifold, is very useful to model vectors that represent true probabilities by construction. Possible applications include probabilistic mixing models and other variations like Bayesian tracer mixing models, which are common in ecology, geoscience, zoology and many others (Phillips, 2012; Stock et al., 2018), and probabilistic treatment of archetype models (Seth and Eugster, 2016; Keller et al., 2021). In some Bayesian settings, level sets of posteriors with one-parameter priors (e.g. sparsity parameter) define star-like manifolds. As we will argue, defining the posterior on such manifolds allows for an objective Bayesian treatment of the problem.

We showcase the generality of our approach with two very distinct applications in Bayesian modeling. First, we introduce a novel Objective Bayesian approach to penalized likelihood methods. In this case the star-like manifold defines a level-set of the penalty constraint. Second, we consider probabilistic mixing models and introduce a general framework for variational inference on the mixing weights. We constrain the posterior on the simplex, such that mixing weights always sum to one, and showcase an application in portfolio analysis.

We summarize the contributions of the present work as follows:

- We propose injective flows for star-like manifolds.
- In contrast to most existing work on injective flows, we show that we can exactly and efficiently compute the associated Jacobian determinant.
- We showcase the relevance and broad applicability of the proposed method in a novel Objective Bayesian approach and for posterior inference in probabilistic mixing models.

## 2 BACKGROUND

**Density and Jacobian determinant for bijective functions** Let $\boldsymbol{x}$ be a $d$-dimensional random variable with unknown distribution $p_x(\boldsymbol{x})$ and let $\boldsymbol{z}$ be a $d$-dimensional random variable with known base distribution $p_z(\boldsymbol{z})$. The key idea of NFs is to model the unknown distribution $p_x(\boldsymbol{x})$ through a transformation $\mathcal{T} : \mathbb{R}^d \mapsto \mathbb{R}^d$ such that $\boldsymbol{x} = \mathcal{T}(\boldsymbol{z})$. If $\mathcal{T}$ is a diffeomorphism, i.e. differentiable bijection with differentiable inverse $\mathcal{T}^{-1}$, the change of variable formula (Rudin, 1987) allows us to express $p_x(\boldsymbol{x})$ solely in terms of the base distribution $p_z(\boldsymbol{z})$ and $\mathcal{T}$: $p_x(\boldsymbol{x}) = p_z(\boldsymbol{z}) \left| \det J_{\mathcal{T}}(\boldsymbol{z}) \right|^{-1}$, where $J_{\mathcal{T}}$ is the Jacobian of the transformation $\mathcal{T}$. The hypothesis can be relaxed to $\mathcal{T}$ being a diffeomorphism a. e. (Munkres and Spivak, 1965), i.e. except on zero-measure sets. Therefore, the trade-off consists of implementing bijections with tractable $\det J_{\mathcal{T}}$ which are still flexible enough to approximate any well-behaved distribution. One key idea is to exploit the property that, given a set of bijections $\{\mathcal{T}^{(i)}\}_{i=1}^k$, their composition $\mathcal{T} = \mathcal{T}^{(k)} \circ \cdots \circ \mathcal{T}^{(1)}$ is still a bijection. Since for bijections the Jacobian is a square matrix, the determinant of a composition of bijections factorizes as the product of the determinants of the individual bijections. Overall, NFs are built as

$$p_x(\boldsymbol{x}) = p_z(\boldsymbol{z}) \left| \det J_{\mathcal{T}}(\boldsymbol{z}) \right|^{-1} \quad \text{with} \quad \det J_{\mathcal{T}}(\boldsymbol{z}) = \prod_{i=1}^k \det J_{\mathcal{T}^{(i)}}(\boldsymbol{u}_{i-1}) \tag{1}$$

where $\boldsymbol{u}_{i-1} = \mathcal{T}^{(i-1)}(\boldsymbol{u}_{i-2})$ and $\boldsymbol{u}_0 = \boldsymbol{z}$. In general, computing the $\det J_{\mathcal{T}}$ has a cost of $O(d^3)$ because it requires some form of matrix decomposition (typically LU or QR). Instead, what most implementation exploit is the property in Eq. (1): we can efficiently model a bijection $\mathcal{T}$ by stacking simpler bijective layers $\mathcal{T}^{(i)}$ with tractable (analytical) Jacobian determinant. Typically, each $\det J_{\mathcal{T}^{(i)}}$ is made tractable by designing bijections $\mathcal{T}^{(i)}$ with triangular Jacobian by construction, such that its determinant is given by the diagonal entries, which has a $O(d)$ complexity.

**Density and Jacobian determinant for injective functions** NFs are limited by the use of bijections, which prevents modeling densities on lower dimensional manifolds. In such cases the target distribution lives on a $m$-dimensional manifold $\mathcal{M}$ embedded in a $d$-dimensional Euclidean space $\mathcal{M} \subset \mathbb{R}^d$, where $m < d$. In order to constrain $p_x(\boldsymbol{x})$ to live on the manifold $\mathcal{M}$, we need an injective transformation that inflates the dimensionality $\mathcal{T}_{\mathrm{m \to d}} : \mathbb{R}^m \mapsto \mathbb{R}^d$. The transformed probability distribution $p_x(\boldsymbol{x})$ can still be computed by accounting for the volume change (Ben-Israel, 1999):

$$p_x(\boldsymbol{x}) = p_z(\boldsymbol{z}) \left| \operatorname{vol} J_{\mathcal{T}_{\mathrm{m \to d}}}(\boldsymbol{z}) \right|^{-1} \quad \text{with} \quad \operatorname{vol} J_{\mathcal{T}_{\mathrm{m \to d}}} = \sqrt{\det \left( \left( J_{\mathcal{T}_{\mathrm{m \to d}}} \right)^T J_{\mathcal{T}_{\mathrm{m \to d}}} \right)}, \tag{2}$$

where $J_{\mathcal{T}_{\mathrm{m \to d}}(\boldsymbol{z})} \in \mathbb{R}^{d \times m}$ is a rectangular matrix. Note that if $m = d$, $J_{\mathcal{T}}$ is a square matrix so $\operatorname{vol} J_{\mathcal{T}} = \sqrt{\det J_{\mathcal{T}}^T J_{\mathcal{T}}} = \sqrt{\det J_{\mathcal{T}}^T \det J_{\mathcal{T}}} = \det J_{\mathcal{T}}$ and Eq. (2) reduces to Eq. (1). Crucially, since $J_{\mathcal{T}_{\mathrm{m \to d}}}$ is now rectangular, the Jacobian determinant cannot be decomposed as the product of stacked transformations anymore, which is the crucial property that makes bijective flows tractable (see Eq. (1)). Instead, we need to explicitly compute the Jacobian determinant, which is $O(m^3)$ in general. This makes injective flows computationally prohibitive for high dimensional manifolds.

Even if we were to stack bijective layers after the injective transformation, we would still be unable to write the overall Jacobian determinants as the product of individual bijections and would remain $O(m^3)$. To see this, consider the transformation $\mathcal{T} = \mathcal{T}_d \circ \mathcal{T}_{m \to d} \circ \mathcal{T}_m$, where $\mathcal{T}_m : \mathbb{R}^m \mapsto \mathbb{R}^m$ and $\mathcal{T}_d : \mathbb{R}^d \mapsto \mathbb{R}^d$ are arbitrary bijections. Then, $\det J_{\mathcal{T}}$ factorizes:

$$\det J_{\mathcal{T}} = \det J_{\mathcal{T}_m} \sqrt{\det \left( \left( J_{\mathcal{T}_d} J_{\mathcal{T}_{m \to d}} \right)^T J_{\mathcal{T}_d} J_{\mathcal{T}_{m \to d}} \right)} . \tag{3}$$

We refer to Appendix A.2 for a full derivation. Note that we can factorize only $\det J_{\mathcal{T}_m}$ (the bijection that precedes the inflating step), while the Jacobian determinant of the bijections $\mathcal{T}_d$ after the inflating step cannot be disentangled. Naturally, the same would hold true if the bijections $\mathcal{T}_d$ were replaced by arbitrary injective transformations. In both cases we would need to compute the Jacobian product $J_{\mathcal{T}_d} J_{\mathcal{T}_{m \to d}}$ and its determinant, which has cubic complexity.

## 3 INJECTIVE FLOWS FOR STAR-LIKE MANIFOLDS

We now present the proposed injective flow to model densities on arbitrary star-like manifolds. In particular, in Section 3.1 we introduce the definition of star-like manifolds and show that they can always be parametrized with generalized spherical coordinates. Then, in Section 3.2 we define injective flows on star-like manifolds and show that the Jacobian determinant can be computed exactly and efficiently. Lastly, in Section 3.3 we discuss the limitations of the proposed approach.

### 3.1 STAR-LIKE MANIFOLDS

**Definition 1.** We call a domain a *star domain* $\mathcal{S}$ if there exists one point $s_0 \in \mathcal{S}$ such that, given any other point $s \in \mathcal{S}$ in the domain, the line segment connecting $s_0$ to $s$ lies entirely in $\mathcal{S}$. Furthermore, we define *star-like manifold* $\mathcal{M}_{\mathcal{S}}$ as the manifold defined by the boundary of a star domain. In this work we consider $d - 1$ dimensional star-like manifolds embedded in $\mathbb{R}^d$.

In other words, a star-like manifold is such there exists a point from which the entire manifold can be "viewed". Intuitively, this suggests that we can always parametrize it with generalized spherical coordinates, which consist of $d - 1$ angles $\boldsymbol{\theta} \in U_{\theta}^{d-1} := [0, \pi]^{d-2} \times [0, 2\pi]$ and one radius $r \in \mathbb{R}_{>0}$. Let $\mathcal{M}_{\mathcal{S}}$ be a $d - 1$ dimensional star-like manifold embedded in $\mathbb{R}^d$. Then, we need $d - 1$ variables to identify any point $\boldsymbol{x} \in \mathcal{M}_{\mathcal{S}}$. In particular, we can parametrize $\boldsymbol{x} = [\boldsymbol{\theta}, r(\boldsymbol{\theta})]^T$ with $d - 1$ spherical angles $\boldsymbol{\theta} \in U_{\theta}^{d-1}$ and a suitable radius function $r(\boldsymbol{\theta})$. If we choose $s_0$ as the origin of the spherical coordinate system, we can define the radius as the line segment connecting $\boldsymbol{x}$ and $s_0$. Crucially, by definition of star-like manifolds, the segment intersects the manifold only once, so the radius is uniquely defined. See Figure 1 for a graphical representation. Star-like manifolds are the most general class of manifolds that always allow such parametrization.

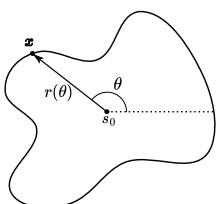

Figure 1: 1D star-like manifold parametrized in spherical coordinates.

### 3.2 PROPOSED INJECTIVE FLOWS FOR STAR-LIKE MANIFOLDS

We now show how to define injective flows on star-like manifolds such that the Jacobian determinant, and hence the log density, can be evaluated exactly and efficiently. Relevantly, the modeled distribution is expressed in Cartesian coordinates, which is crucial for most applications. In order to do so we compose three transformations $\mathcal{T} := \mathcal{T}_{s \to c} \circ \mathcal{T}_r \circ \mathcal{T}_{\theta}$ (see Figure 2): (i) an arbitrary diffeomorphism that maps to $d - 1$ spherical angles $\mathcal{T}_{\theta}$, (ii) the injective transformation $\mathcal{T}_r$ that parametrizes the radius $r(\boldsymbol{\theta})$ as a function of the angles and (iii) the coordinate transformation $\mathcal{T}_{s \to c}$ from spherical to Cartesian coordinates. Note that the change of coordinates $\mathcal{T}_{s \to c}$ is a diffeomorphism almost everywhere, i.e. except on a zero-measure set. Therefore, the change of variable formula holds and the probability resulting from the transformation is still exact. We refer to Appendix A.1 and A.4 for the explicit expression of $\mathcal{T}_{s \to c}$ and a discussion of its special structure, respectively. This way the resulting densities will be defined by construction on the manifold parametrized by $\mathcal{T}_r$, while the density will be conveniently expressed in Cartesian coordinates. In other words, the push-forward samples are points in the ambient space.

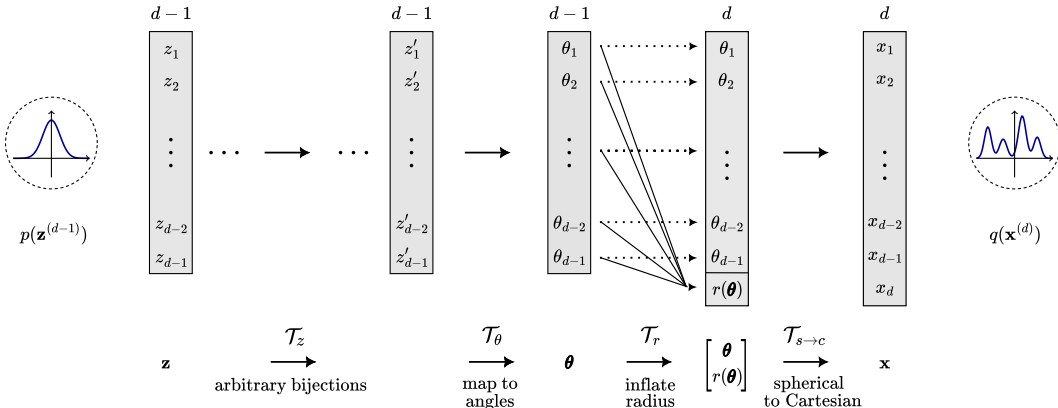

Figure 2: Architecture of the proposed injective flows for star-like manifolds, see Theorem 1.

Naively computing the Jacobian determinant as in Eq. (3) requires cubic complexity. Instead, in Theorem 1 we show for star-like manifolds how to compute it analytically and efficiently in $O(d^2)$. We provide a proof sketch of the theorem below and the full proof in the Appendix A.4.

**Theorem 1.** *Let* $\mathcal{T} := \mathcal{T}_{s \to c} \circ \mathcal{T}_r \circ \mathcal{T}_\theta$ *as in Figure 2, where* $\mathcal{T}_\theta : z \in \mathbb{R}^{d-1} \mapsto \theta \in U_\theta^{d-1}$ *is any diffeomorphism to* $d$*-spherical angles,* $\mathcal{T}_r : \theta \in U_\theta^{d-1} \mapsto [\theta, r(\theta)] \in U_\theta^{d-1} \times \mathbb{R}_{>0}$ *with* $r(\cdot) : \theta \in U_\theta^{d-1} \mapsto r \in \mathbb{R}_{>0}$ *being differentiable, and* $\mathcal{T}_{s \to c} : [\theta, r(\theta)]^T \in U_\theta^{d-1} \times \mathbb{R}_{>0} \mapsto x \in \mathbb{R}^d$ *the* $d$*-spherical to Cartesian transformation (see Definition 2 in Appendix A.1).*

*Then, the Jacobian determinant of the full transformation* $\mathcal{T}$ *is equal to*

$$\det\left(J_\mathcal{T}\right) = \det\left(J_{\mathcal{T}_\theta}\right) \det\left(J_{\mathcal{T}_{s \to c}}\right) \left\|\left(J_{\mathcal{T}_{s \to c}}^T\right)^{-1} y\right\|_2, \tag{4}$$

*where* $y := \left[-\nabla_\theta r(\theta), 1\right]^T$*. Relevantly,* $\det J_\mathcal{T}$ *can be computed exactly and efficiently in* $O(d^2)$*.*

*Proof sketch. (i) Injectivity* It is easy to see that $\mathcal{T} := \mathcal{T}_{s \to c} \circ \mathcal{T}_r \circ \mathcal{T}_\theta$ is injective because $\mathcal{T}_\theta$ is bijective, $\mathcal{T}_r$ is injective and $\mathcal{T}_{s \to c}$ is also bijective. The injectivity of $\mathcal{T}_r$ can be easily seen by noting that $\theta \neq \theta' \implies [\theta, r(\theta)] \neq [\theta', r(\theta')]$, independently of $r(\cdot)$.

*(ii) Analytical Jacobian determinant* Since $J_\mathcal{T} \in \mathbb{R}^{d \times d-1}$ is not square, we cannot use the usual property of bijections in Eq. (1). Instead, we use the result for injective functions in Eq. (3):

$$\det\left(J_\mathcal{T}\right) = \det\left(J_{\mathcal{T}_\theta}\right) \det\left(J_{\mathcal{T}_{s \to c} \circ \mathcal{T}_r}\right) = \det\left(J_{\mathcal{T}_\theta}\right) \sqrt{\det\left(\left(J_{\mathcal{T}_{s \to c}} J_{\mathcal{T}_r}\right)^T \left(J_{\mathcal{T}_{s \to c}} J_{\mathcal{T}_r}\right)\right)}.$$

where $J_{\mathcal{T}_r} \in \mathbb{R}^{d \times d-1}$ and $J_{\mathcal{T}_{s \to c}} \in \mathbb{R}^{d \times d}$. Note that, as we show in Appendix A.4 Remark 1, we can factor out only the Jacobian determinant of $\mathcal{T}_\theta$, i.e. the bijection that precedes the dimensional inflation step with $\mathcal{T}_r$. The term $\det J_{\mathcal{T}_\theta}$ is the standard Jacobian determinant for bijective layers and can be computed efficiently. We are then left to compute $\det\left(J_{\mathcal{T}_{s \to c} \circ \mathcal{T}_r}\right)$. To do so we consider the matrix $\tilde{J}_{\mathcal{T}_r} := [J_{\mathcal{T}_r} \; \mathbf{0}_{d \times 1}] \in \mathbb{R}^{d \times d}$ and substitute the determinant with the pseudo-determinant:

$$\det\left(J_{\mathcal{T}_{s \to c} \circ \mathcal{T}_r}\right) = \sqrt{\det\left(J_{\mathcal{T}_r}^T J^* J_{\mathcal{T}_r}\right)} = \sqrt{\det{}^+\left(\tilde{J}_{\mathcal{T}_r}^T J^* \tilde{J}_{\mathcal{T}_r}\right)},$$

where $J^* := J_{\mathcal{T}_{s \to c}}^T J_{\mathcal{T}_{s \to c}} \in \mathbb{R}^{d \times d}$. With $\det{}^+$ we denote the pseudo-determinant, which is defined as the product of all non-zero eigenvalues. The rest of the proof is based on the key observation that $\tilde{J}_{\mathcal{T}_r}$ has rank $d-1$ or, equivalently, that its null space is one-dimensional. As a result, we can use Lemma 2 (see Appendix A.4) and re-write the pseudo-determinant as trace of the adjugate matrix :

$$\det{}^+\left(\tilde{J}_{\mathcal{T}_r}^T J^* \tilde{J}_{\mathcal{T}_r}\right) = \text{Tr}\left(\text{adj}\left(\tilde{J}_{\mathcal{T}_r}^T J^* \tilde{J}_{\mathcal{T}_r}\right)\right) = \det\left(J^*\right) \text{Tr}\left(\text{adj}\left(\tilde{J}_{\mathcal{T}_r}\right)\left(J^*\right)^{-1} \text{adj}\left(\tilde{J}_{\mathcal{T}_r}^T\right)\right).$$

where $\text{adj}(\cdot)$ is defined as the transpose of the cofactor matrix. If $A$ is invertible (as for $J^* = J_{\mathcal{T}_{s \to c}}^T J_{\mathcal{T}_{s \to c}}$), $\text{adj}(A) = \det(A) A^{-1}$. Since $\tilde{J}_{\mathcal{T}_r}^T$ has rank $d-1$, Lemma 1 holds (see Appendix A.4):

$$\text{adj}(\tilde{J}_{\mathcal{T}_r}) = \frac{\det{}^+(\tilde{J}_{\mathcal{T}_r})}{y^T x} x y^T,$$

where $x = [\mathbf{0}, 1]^T$ and $y := \big[ -\nabla_{\boldsymbol{\theta}} r(\boldsymbol{\theta}), 1 \big]^T$. Finally, by substitution we get:

$$\det{}^+\left( \tilde{J}_{\mathcal{T}_r}^T J^* \tilde{J}_{\mathcal{T}_r} \right) = \det\left( J^* \right) \frac{\det{}^+(\tilde{J}_{\mathcal{T}_r})^2}{(y^T x)^2} \operatorname{Tr}\left( xy^T \left( J^* \right)^{-1} yx^T \right) = \det\left( J_{\mathcal{T}_{s \to c}} \right)^2 \| \left( J_{\mathcal{T}_{s \to c}}^T \right)^{-1} y \|_2^2$$

*(iii) Complexity* We can now analyze the time complexity required to evaluate Eq. (4). The term $\det J_{\mathcal{T}_\theta}$ is the standard Jacobian determinant for bijective layers, which can be computed in $O(d)$. The Jacobian determinant for spherical to Cartesian coordinates $\det J_{s \to c}$ is known in closed form (Muleshkov and Nguyen, 2016) and can be evaluated in $O(d)$. Therefore, we only need to show that also $w = \left( J_{\mathcal{T}_{s \to c}}^T \right)^{-1} y$ can be computed efficiently. Solving the full linear system would require a complexity of $O(d^3)$. However, $J_{\mathcal{T}_{s \to c}}^T$ is almost-triangular (see Appendix A.4, Eq. (22)) and we can make it triangular with one step of Gaussian elimination, which requires $O(d^2)$. The resulting triangular system can be solved in $O(d^2)$. Finally, note that we can compute $J_{\mathcal{T}_{s \to c}}$ very efficiently and analytically (see Appendix A.4, Eq. (24)), without requiring autograd computations. ■

### 3.3 FURTHER DETAILS AND LIMITATIONS

**Variational inference vs maximum likelihood**    In this paper we focus on variational inference settings (without observations), and not on the maximum likelihood setting (with observations), which is the main focus of most injective flows. Our motivation is two-fold. First, our method is designed for star-like manifolds (with codimension 1), which are more useful in variational inference settings. In maximum likelihood applications, the underlying manifolds are often assumed to much lower dimensional. Second, while in maximum likelihood settings the approximate Jacobian work well in practice, in variational inference settings the exact Jacobian determinant is crucial to learn the correct target distribution. We show this empirically, already in very simple cases.

**Implementation details**    The proposed approach is straightforward to implement and only requires to adapt the function $r(\boldsymbol{\theta})$ to the star-like manifold under study. This means that we should have access to the parametrization $r(\boldsymbol{\theta})$, which is often trivial (see e.g. Appendix A.4 Eq. (25) and Eq. (26)). Our method is also versatile as the transformation $\mathcal{T}_z$ can be implemented with any bijective layers of choice. Lastly, we easily avoid numerical instabilities arising from singular points in $\mathcal{T}_{s \to c}$ by offsetting the critical angles with a small epsilon.

**Limitations**    The main limitation of the proposed method is that it cannot be easily generalized to manifolds with intrinsic dimensionality lower than $d - 1$. In particular, the proof of Theorem (1) relies on the fact that the zero-padded Jacobian matrix has rank $d - 1$. Secondly, the expressivity of the proposed method also depends on the flexibility of the bijective layers. Despite state-of-the-art bijective layers being extremely expressive (Perugachi-Diaz et al., 2021), Liao and He (2021) showed that the number of modes that can be modeled remains limited.

## 4 RELATED WORK

**Normalizing Flows**    Normalizing Flows (NFs) consist of a simple base distribution that is transformed into a more complicated one through bijective transformations. One can show that such a construction allows us to approximate any well-behaved distribution (Papamakarios et al., 2021). In practice, the bijective transformations are implemented with neural networks that show a trade-off between expressiveness and computational complexity. However, recently developed bijective layers provide very efficient transformations that satisfy the universality property (Huang et al., 2018; Durkan et al., 2019; Jaini et al., 2019). For a comprehensive review of different bijective layers and for a discussion about applications we refer to Papamakarios et al. (2021) and Kobyzev et al. (2021).

**Variational Inference with NFs**    NFs are popular in two distinct applications: (i) as generative models trained on data or (ii) as powerful density estimators in variational inference settings. In the first case, we are given some samples and NFs are trained by maximum likelihood (forward KL divergence) to approximate the data generating distribution and to later generate new samples (Dinh et al., 2017; Papamakarios et al., 2017). In the variational inference setting, no samples are available and the target distribution is typically known only up to a normalization constant. NFs are thus

trained by reverse KL divergence and, once trained, they allow for both sampling and evaluation of the (approximate) target distribution (Rezende and Mohamed, 2015; Kingma et al., 2016). There is good evidence that a more faithful posterior approximation leads to an improved performance for variational inference tasks (Rezende and Mohamed, 2015), which is one reason that makes normalizing flows attractive. This setting is the focus of our work and is particularly relevant for Bayesian inference (Louizos and Welling, 2017), where the goal is to learn and sample from the posterior distribution given the (unnormalized) product of the prior and likelihood. In such settings NFs have also proven to be an attractive alternative to MCMC samplers (Negri et al., 2023).

**Injective flows on manifolds** The computational challenge of injective flows is the evaluation of the Jacobian determinant in Eq. (2). In the literature, exact and efficient computation of the Jacobian determinant has been shown only for trivial manifolds like spheres and tori (Gemici et al., 2016; Rezende et al., 2020) or for very restrictive transformations (Ross and Cresswell, 2021). In particular, the latter focuses only on maps for which $\mathcal{J}^T \mathcal{J} \propto \mathbb{I}$, making the determinant trivial. These are however the only cases where the modeled density can be computed exactly and efficiently during training. This has a significant impact on variational inference settings where the quality of the approximation influences the exploration of the distribution domain. It is, in contrast, less relevant in data driven settings trained by maximum likelihood. Some early work ignored the determinant term altogether (Brehmer and Cranmer, 2020), which was shown to have detrimental effects already in simple low-dimensional settings (Caterini et al., 2021). Current work is focused instead on finding some tractable approximation to the Jacobian determinant. The most common one is to employ the Hutchinson's trace estimator (Mathieu and Nickel, 2020; Caterini et al., 2021; Flouris and Konukoglu, 2023), which is characterized by high variance and is biased if used to estimate the log-determinant of the Jacobian (Kumar et al., 2020). State-of-the-art works employ surrogate log-likelihood losses and still approximate the Jacobian determinant (Sorrenson et al., 2024).

In contrast, we are the first to propose exact and computationally efficient injective flows for a wide class of manifolds, namely star-like manifolds.

## 5 APPLICATIONS

In this section we showcase some interesting applications of the proposed injective flows. In particular, we overcome some limitations of current work with exact density estimation, which are applicable only to trivial manifolds or for very restrictive transformations. In Section 5.1 we show empirically that the proposed approach provides a significant speedup compared to the explicit computation of the Jacobian determinant. We also show that in variational inference the exact Jacobian determinant is crucial for training while the approximation commonly employed in injective flows results in poor reconstruction. In Section 5.2 we illustrate how the proposed approach learns distributions on simple 3D manifolds. In Section 5.3 we use injective flows to define a novel Objective Bayes approach to penalized likelihood problems. Lastly, in Section 5.4 we introduce a general framework for variational inference in probabilistic mixing models.

### 5.1 EFFECTIVENESS AND EFFICIENCY OF THE PROPOSED METHOD

We showed that for star-like manifolds we can efficiently compute the Jacobian determinant in Eq. (2) and we argued that exact evaluation is crucial for variational inference. We now verify both statements empirically. Firstly, we compare the runtime associated with computing the Jacobian determinant via the explicit formula in Eq. (2) with our approach in Eq. (4). In particular, we use a very simple manifold, the hypersphere in $d$ dimensions, and measure runtime over 20 repetitions of Jacobian determinant computation. The results in Figure 3a show that our method provides a significant speedup compared to the explicit computation of the Jacobian. By fitting a linear function to the log-runtime and log-dimension we get that the explicit computation is approximately cubic, $O(d^{2.96})$, while our proposed approach is approximately quadratic, $O(d^{1.81})$. Secondly, we show that training with the exact Jacobian determinant is crucial for variational inference. To do so we first train the proposed injective flow with the exact (and efficient) Jacobian in Eq. (4). Then, we train a second identical flow where the gradient of the Jacobian is approximated with the Hutchinson trace estimator with different number of Gaussian samples $n = 1, 10, 50, 100$. This approach is common in most injective flow papers (Caterini et al., 2021; Flouris and Konukoglu, 2023). We use

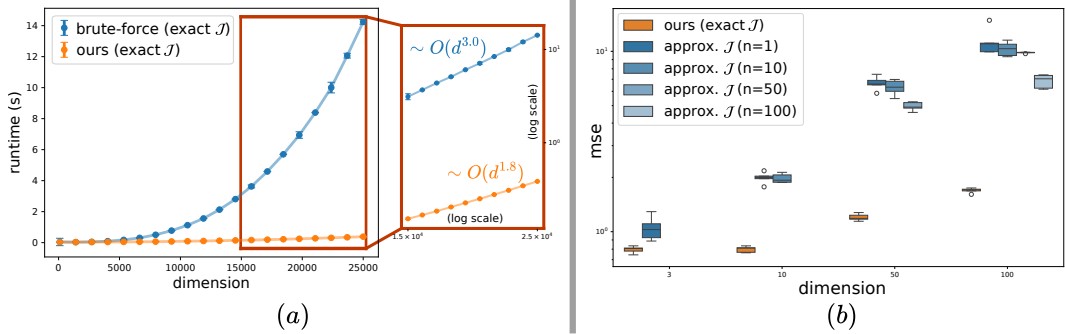

$(a)$ $(b)$

Figure 3: (a) Runtime comparison of Jacobian determinant computation with proposed approach in Eq (4) and with the brute-force computation Eq. (2). (b) MSE of learned log-density for an injective flow trained to learn a uniform distribution on a $l_p$-(pseudo) norm ball with $p = 0.5$. We compare the same model trained with our exact Jacobian and with the Hutchinson trace estimator with $n = 1, 10, 50, 100$ number of Gaussian samples. Note that the number of samples is upper bounded by the dimensionality of the problem and that at evaluation the exact Jacobian is used.

the implementation of state-of-the-art work (Sorrenson et al., 2024). The task is to learn the uniform distribution on a lp-(pseudo) norm ball with $p = 0.5$. In Figure 3 we compare the ground truth (log-) density with that obtained with the two models. Note that at test time the model trained with the approximate Jacobian is evaluated with the exact Jacobian. In the Appendix in Figure 10 we measure the similarity between samples of the two models and samples from the true distribution. In all cases the proposed model achieves significantly better results both in terms of density reconstruction and samples quality. Even at the cost of increased runtime (shown in Figure 10), increasing the number of samples the approximate method fails to achieve similar performance to ours.

## 5.2 ILLUSTRATIVE DISTRIBUTIONS IN 3D

We first illustrate the proposed model on a simple 3D setting. In particular, we train the injective flow in a variational inference setting where the target distribution is known only up to a constant and no samples are given. Note that this is much more challenging than the usual maximum likelihood settings where flows are trained on data. In particular, it is not trivial to learn densities with many modes. To showcase the effectiveness of the proposed method we show the modeled density for increasingly difficult targets: (i) von Mises-Fisher distribution ($\kappa = 5$), (ii) mixture of 50 von Mises-Fisher distributions arranged on a spiral ($\kappa = 50$) and (iii) sinusoidal density ($\log \rho(\theta, \phi) = \sin(4\theta)\sin(4\phi)$), where $\kappa$ is the concentration parameter and $\theta$ and $\phi$ are the polar and azimuthal angles, respectively. The sinusoidal density (iii) is defined on a deformed sphere. In Figure 4 we show that the modeled densities perfectly match the ground truth in all cases. Furthermore, we compare the modeled density with the ground truth over 10'000 samples and obtained an accurate reconstruction: MSE $= 0.013$ (a), $0.011$ (b). As the normalized density on the deformed sphere is not given, we only provide a qualitative visualization, which shows an accurate reconstruction.

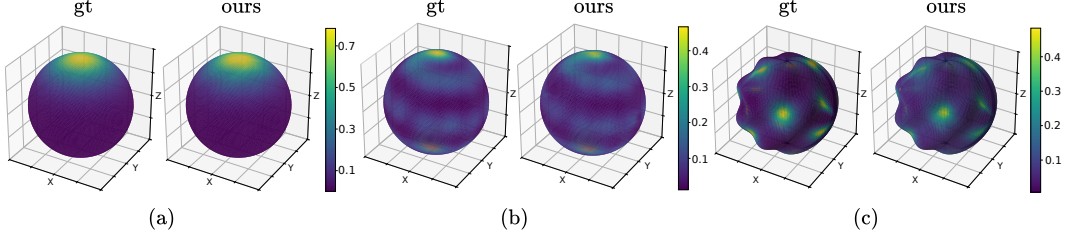

$(a)$ $(b)$ $(c)$

Figure 4: Learned ("ours") and ground truth ("gt") density for proposed injective flow trained via reverse KL divergence (no samples). Only the (unnormalized) target is given: (a) von Mises-Fisher ($\kappa = 5$), (b) mixture of 50 von Mises-Fisher ($\kappa = 50$) and (c) sinusoidal density on deformed sphere .

### 5.3 Objective Bayesian approach to penalized likelihood

**Objective and subjective Bayes**   Bayesian inference is a powerful statistical method that requires a likelihood term, which explains the observed data, and a prior, which quantifies our initial belief. However, in many cases we might not have enough problem-specific knowledge to specify an informative subjective prior. This has led to the development of *objective priors*, which are designed to be minimally informative. Some objective priors include Jeffreys rule (Jeffreys, 1961), reference priors (Bernardo, 1979), and maximum entropy priors (Jaynes, 2003). Given the vast literature on objective priors (Berger, 2006), in this work we do not intend to discuss whether objective priors are preferable or not. Instead, we provide a new framework to define objective priors in settings where only subjective ones have been explored so far. Specifically, we consider Bayesian penalized likelihood problems and show that level sets of the penalty define star-like manifolds. Then, we implicitly define the objective prior as the uniform distribution on such manifolds. This choice aligns with the literature on objective priors for distributions on surfaces since a uniform distribution assigns equal mass to equal volume (Kass, 1989; Kass and Wasserman, 1996).

**Objective Bayes for penalized likelihood models**   Let $y \sim X\beta + \epsilon$ with $\epsilon \sim \mathcal{N}(0, \sigma^2 \mathbb{I}_n)$, where $X \in \mathbb{R}^{n \times d}$ is the data matrix, $y \in \mathbb{R}^n$ the targets and $\beta \in \mathbb{R}^d$ the regression coefficients. We then optimize the mean-squared error $\|y - X\beta\|_2^2$ subject to the (pseudo-) norm penalties $\|\beta\|_p^p$ with $p > 0$, which encourages sparsity for $p \leq 1$. Note that for $p = 1$ we recover the LASSO penalty (Tibshirani, 1996) and for $p = 2$ the Ridge penalty. Tibshirani (1996) noted that we can interpret such penalized likelihood in a Bayesian way with a Gaussian likelihood and a suitable prior. Park and Casella (2008) showed that with an independent Laplace prior the Maximum a Posteriori (MAP) of the posterior coincides with the frequentist solution. The above reasoning can be extended to any $l_p$ (pseudo-) norm $\|\cdot\|_p$ by using a generalized Gaussian prior on $\beta$:

$$\operatorname*{arg\,min}_{\beta \in \mathbb{R}^d} \tfrac{1}{2\sigma^2}\|y - X\beta\|_2^2 + \lambda\|\beta\|_p^p = \operatorname*{arg\,max}_{\beta \in \mathbb{R}^d} \underbrace{\mathcal{N}(X\beta, \sigma^2\mathbb{I}_n)}_{p(y|X,\beta)} \underbrace{\prod_i \exp\{-\lambda|\beta_i|^p\}}_{p(\beta|\lambda)} = \beta^* . \quad (5)$$

However, the generalized Gaussian is not the only prior for which Eq. (5) holds. Any monotonic transformation $h$ of $p(\beta|\lambda)$ results in the same MAP solution $\beta^*$ (see Figure 8 in the Appendix). Therefore, the choice of $h(p(\beta|\lambda))$, and hence of the prior, is subjective and, crucially, it influences the posterior. We show this empirically on toy data with the Laplace prior and two monotonic transformations: the square ("square laplace") and square root ("root laplace"). In Figure 5 we can clearly see that the posterior is influenced by the choice of the subjective prior, which is undesirable in the absence of specific assumptions. In contrast, we circumvent the choice of a subjective prior and propose a general framework for objective priors in penalized likelihood methods.

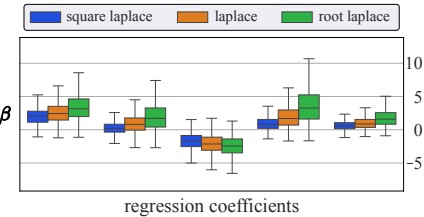

Figure 5: 95% posterior C.I. for 3 subjective priors with the same MAP. The choice of prior affects the posterior.

**Objective Bayesian penalized likelihood with injective flows**   With subjective priors the penalty $\|\beta\|_p^p$ in Eq. (5) is enforced as a soft constraint controlled by $\lambda$ such that $\|\beta\|_p \leq k(\lambda)$, for some $k(\lambda)$. Our idea is to enforce the norm penalty as a hard constraint by defining the posterior on the manifold $\|\beta\|_p = k$ by construction. This way we do not require to *explicitly* specify a subjective prior and we are *implicitly* assuming a uniform prior on the manifold $\|\beta\|_p = k$. In summary:

| *Objective Bayes* | | *Subjective Bayes* |
|---|---|---|
| $p(y\|X,\beta) = \mathcal{N}(X\beta, \sigma^2\mathbb{I}_n)$ | $\longleftrightarrow$ | $p(y\|X,\beta) = \mathcal{N}(X\beta, \sigma^2\mathbb{I}_n)$ |
| posterior on manifold: $\|\beta\|_p = k$ | | prior: $p(\beta\|\lambda) \propto \prod_i \exp\{-\lambda\|\beta_i\|^p\}$ |

The equality $\|\beta\|_p = k$ induces a star-like manifold which we can parametrize with a suitable radius function; see Appendix A.5 Eq. (25) for the explicit parametrization. Therefore, with our framework we can define the (approximate) posterior $q_\theta(\beta)$ to be constrained on $\|\beta\|_p = k$ by construction.

We now illustrate the differences between the subjective and objective approaches with synthetic data. We use a NF to approximate the posterior $\mathcal{N}(\boldsymbol{X}\boldsymbol{\beta}\sigma^2\mathbb{I}_n)p(\boldsymbol{\beta}|\lambda)$ with the "square laplace" subjective prior $p(\boldsymbol{\beta}|\lambda)$. We choose $\lambda$ such that the MAP has a specific norm $\|\boldsymbol{\beta}^*\|_1 = k$. Furthermore, we use an injective flow defined on $\|\boldsymbol{\beta}\|_1 = k$ to approximate the target $\mathcal{N}(\boldsymbol{X}\boldsymbol{\beta}, \sigma^2\mathbb{I}_n)$ (implicitly uniform prior). In both cases training is performed by minimizing the reverse KL divergence (see Appendix A.6, Eq.(27)). Figure 6 shows a crucial difference: samples from the objective posterior lie exactly on the manifold while the subjective posterior is scattered. The bottom panel of Figure 6 shows the distribution of the sample norms varying significantly with the choice of the subjective prior, which agrees with Figure 5. We include

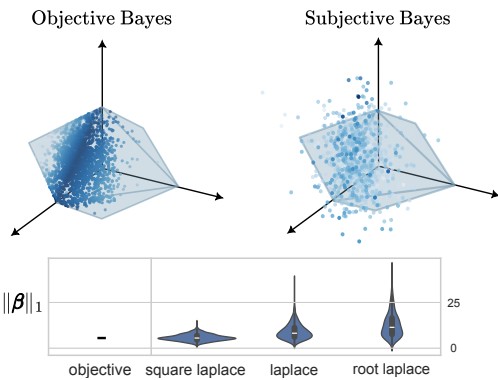

Figure 6: *Above:* with the objective prior posterior samples lie on the manifold. *Below:* the norm of posterior samples depends on the subjective prior.

more implementation details in Appendix A.6. Lastly, note that we do not claim that the proposed prior is *better* in general, but rather preferable in cases where a less informative prior is desired.

## 5.4 VARIATIONAL INFERENCE FOR PROBABILISTIC MIXING MODELS

**Probabilistic mixing models**    Mixing models are used to study the relative contribution of sources to an observed mixture and are particularly relevant in ecology, geoscience, meteorology and many other fields (Phillips, 2012; Stock et al., 2018; Jiskra et al., 2021). Formally, the aim is to reconstruct the mixture $\boldsymbol{\pi}$ of sources from which the observations $\mathcal{D}$ were generated, with $\boldsymbol{\pi}$ being defined on the probabilistic simplex $\mathcal{C}^d := \left\{\boldsymbol{\pi} \in \mathbb{R}^d : \pi_i \geq 0 \; , \; \sum_{i=1}^{k} \pi_i = 1\right\}$. In the most general Bayesian formulation, we require a prior $p(\boldsymbol{\pi})$ and some likelihood $p(\mathcal{D}|\boldsymbol{\pi})$ and the challenge is then to define the posterior $p(\boldsymbol{\pi}|\mathcal{D}) \propto p(\mathcal{D}|\boldsymbol{\pi})p(\boldsymbol{\pi})$ on the probabilistic simplex $\mathcal{C}^d$. Most approaches rely on the Dirichlet distribution, which is defined on $\mathcal{C}^d$ by construction: $\mathrm{Dir}(\boldsymbol{\pi}) \propto \prod_i \pi_i^{\alpha_i - 1}$ with $\alpha_i > 0$. Since the posterior can be obtained in closed form only with a multinomial likelihood, most approaches rely on sophisticated MCMC samplers (Stock et al., 2018). As a more flexible alternative to MCMC methods, we present a general variational inference framework where $p(\boldsymbol{\pi}|\mathcal{D})$ is always defined on $\mathcal{C}^d$, leaving complete freedom in the choice of prior and likelihood.

**Injective flows vs MCMC in Bayesian mixing models**    It is easy to see that the probabilistic simplex $\mathcal{C}^d$ is a star-like manifold, see Appendix A.5 Eq. (26) for the explicit parametrization. Therefore, with the our framework we can define an injective flow $q_{\boldsymbol{\theta}}(\boldsymbol{\pi})$ on $\mathcal{C}^d$ by construction and train it to approximate the posterior $p(\boldsymbol{\pi}|\mathcal{D}) \propto p(\mathcal{D}|\boldsymbol{\pi})p(\boldsymbol{\pi})$. In the simplest case when no prior is specified, we are implicitly assuming a uniform distribution on the simplex, i.e. a Dirichlet prior with $\alpha_i = 1 \; \forall i$. In the more general case, we can always plug in any combination of likelihood $p(\mathcal{D}|\boldsymbol{\pi})$ and prior $p(\boldsymbol{\pi})$, and the (approximate) posterior $q_{\boldsymbol{\theta}}(\boldsymbol{\pi})$ will always be defined on $\mathcal{C}^d$ by design. In contrast, with MCMC methods it is not trivial to guarantee posterior samples to be on $\mathcal{C}^d$, already for very simple likelihoods (Altmann et al., 2014; Baker et al., 2018). We compare our method with an MCMC sampler in the conjugate case (Dirichlet prior and multinomial likelihood) such that we can compare with the true posterior, which is available in closed form. We report the results in Appendix A.6 in Figure 11, together with a description of the MCMC sampler. Results show that (i) the proposed injective flow correctly estimates the posterior distribution across all tested dimensions and (ii) it outperforms the MCMC sampler already in low dimensions. Interestingly, this experiment shows that our approach is able to learn very sparse solutions with many modes.

**Application: Bayesian portfolio optimization**    We select a minimal Bayesian mixing model that already shows the advantages of our proposed method in terms of flexibly choosing likelihood and prior. One such setting is index replication in the context of portfolio optimization (Markowitz and Todd, 2000). A portfolio is defined as a set of $n$ stocks which are held proportionally to the mixture components $\boldsymbol{\pi} \in \mathbb{R}^n_{>0}$, such that $\sum_i \pi_i = 1$. Let $\boldsymbol{R} \in \mathbb{R}^{T \times n}$ be the returns over the time-steps $t = \{1, \dots, T\}$ of the $n$ stocks. We are interested in optimizing the portfolio weights

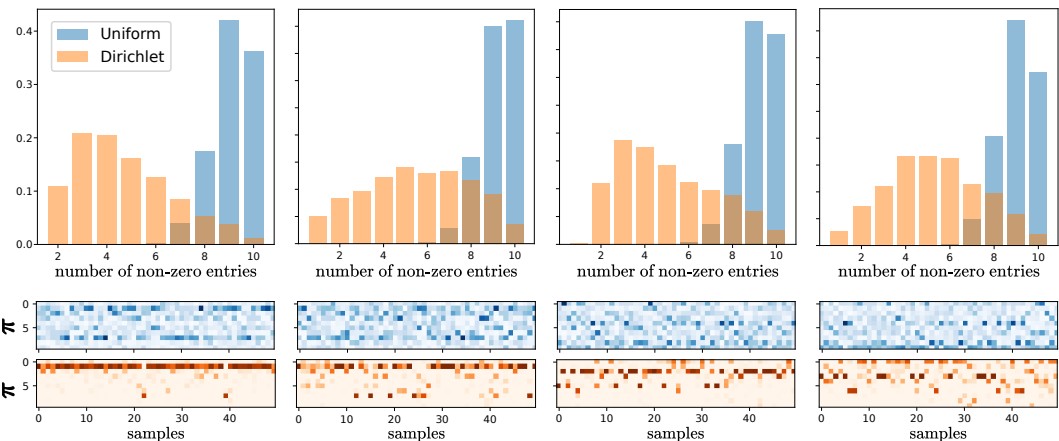

Figure 7: Trained injective flow on the simplex with target $p(\boldsymbol{\pi}|\mathcal{D}) \propto p(\mathcal{D}|\boldsymbol{\pi})$ ("uniform" prior, implicitly) and $p(\boldsymbol{\pi}|\mathcal{D}) \propto p(\mathcal{D}|\boldsymbol{\pi})\mathrm{Dir}(\boldsymbol{\pi})$ ("Dirichlet" prior). *Above*: we compare the distribution of non-zero entries in posterior samples for 4 fixed likelihood values (so at fixed reconstruction). *Below*: we compare sparsity patterns of samples $\boldsymbol{\pi}$ with uniform (blue) and Dirichlet (orange) prior.

$\boldsymbol{\pi}$ such that we replicate the reference index returns $\boldsymbol{\rho} \in \mathbb{R}^T$, while also incorporating investors personal preferences. For instance, a sparse portfolio allows to reduce transaction costs arising from trading (Sokolov and Polson, 2019). We formulate the problem in Bayesian fashion by specifying a Gaussian likelihood $p(\boldsymbol{\rho}|\boldsymbol{R}, \boldsymbol{\pi}) = \mathcal{N}(\boldsymbol{R}\boldsymbol{\pi}, \sigma^2 \mathbb{I}_n)$ and some sparsity-inducing prior $p(\boldsymbol{\pi})$.

With the proposed framework we can approximate the posterior $p(\boldsymbol{\pi}|\boldsymbol{R}, \boldsymbol{\rho}) \propto p(\boldsymbol{\rho}|\boldsymbol{R}, \boldsymbol{\pi})p(\boldsymbol{\pi})$ with an injective flow $q_{\boldsymbol{\theta}}(\boldsymbol{\pi})$ defined on $\mathcal{C}^n$ by design. The flow $q_{\boldsymbol{\theta}}(\boldsymbol{\pi})$ is trained by minimizing the reverse KL divergence with the unnormalized target $p(\boldsymbol{\rho}|\boldsymbol{R}, \boldsymbol{\pi})p(\boldsymbol{\pi})$. For the sake of illustration, we select a portfolio with 10 stocks over a period of 200 time steps from the dataset in Tu and Li (2024). We define $q_{\boldsymbol{\theta}}(\boldsymbol{\pi})$ on the manifold and consider two priors: the uniform prior on the simplex and the Dirichlet distribution. In Figure 7 we show the distribution of non-zero entries of the posterior samples for the uniform and Dirichlet distribution. In particular, we consider the distribution and the sparsity patterns at 4 fixed values of the likelihood (one per plot). Despite the likelihood being the same, the Dirichlet prior leads to a sparser solution with fewer non-zero entries. This is also noticeable in the sparsity patterns of the posterior samples in the bottom panel. In Appendix A.6 in Figure 12 we also show the cumulative return and how it is affected by sparsity. Overall, we showed how easily we can specify any likelihood and priors while constraining the posterior on the simplex.

## 6 CONCLUSIONS

Previous work on injective flows on manifolds relies on approximations or lower bounds to circumvent the computation of the Jacobian determinant term. In this work we showed how to exactly and efficiently compute the Jacobian determinant term for the general class of star-like manifolds. We validate empirically the claimed computational advantage and we show that exact Jacobian computation is crucial for variational inference, already in very simple settings. We then showed that the proposed flow allows for interesting applications that were not possible before. First, with the proposed framework we introduced a novel Objective Bayes approach to penalized likelihood methods. The idea is to circumvent the choice of a subjective prior by constraining the posterior on the manifold defined by level-sets of the prior. Second, we introduced a general variational inference framework for modeling the posterior in probabilistic mixing models. Overall, the proposed framework allows us to efficiently model distributions on arbitrary star-like manifolds and to flexibly specify any choice of prior and likelihood.

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

## A    APPENDIX

The Appendix is organized in six parts. In Subsection A.1 we define the generalized spherical coordinate system and define the transformation from spherical to Cartesian coordinates. In Subsection A.2 and A.3 we provide some auxiliary theorems and Lemmas that are used in the main proof. In Subsection A.4 we provide the full proof of Theorem 1. In Subsection A.5 we provide some details about the implementation of the proposed injective flows and we make some further comments about the associated computational cost. Finally, in Subsection A.6 we include further plots and implementation details about the experiments.

### A.1    GENERALIZED SPHERICAL COORDINATES

**Definition 2.** We define the *d-spherical coordinate system* as a generalization of the spherical coordinate system for $d$-dimensional Euclidean spaces. Such coordinate system is defined with $d - 1$ angles $\theta_1, \ldots, \theta_{d-1}$ and one radius $r \in \mathbb{R}_{>0}$, where $\theta_i \in [0, \pi]$ for $i < d - 1$ and $\theta_{d-1} \in [0, 2\pi]$. We further define a transformation $\mathcal{T}_{s \to c} : \boldsymbol{x}_s \mapsto \boldsymbol{x}_c$ that maps spherical coordinates $\boldsymbol{x}_s = [\theta_1, \ldots, \theta_{d-1}, r]^T$ to Cartesian coordinates $\boldsymbol{x}_c = [x_1, \ldots, x_d]^T$ as

$$
\begin{cases}
x_1 = r \cos \theta_1 \\
x_2 = r \sin \theta_1 \cos \theta_2 \\
\quad \vdots \\
x_{d-1} = r \sin \theta_1 \sin \theta_2 \cdots \sin \theta_{d-2} \cos \theta_{d-1} \\
x_d = r \sin \theta_1 \sin \theta_2 \cdots \sin \theta_{d-2} \sin \theta_{d-1}
\end{cases}
\tag{6}
$$

We denote with $U_\theta^{d-1} \times \mathbb{R}_{>0}$ the domain of definition for $d$-spherical coordinate system, where $U_\theta^{d-1} := [0, \pi]^{d-2} \times [0, 2\pi]$.

### A.2    JACOBIAN DETERMINANT FOR ARBITRARY INJECTIVE FLOWS

**Remark 1.** *Let $\mathcal{T}_m : \mathbb{R}^m \mapsto \mathbb{R}^m$ and $\mathcal{T}_d : \mathbb{R}^d \mapsto \mathbb{R}^d$ be arbitrary bijective transformation and let $\mathcal{T}_{m \to d} : \mathbb{R}^m \to \mathbb{R}^d$ be an injective transformation with $m < d$. The transformation $\mathcal{T} = \mathcal{T}_d \circ \mathcal{T}_{m \to d} \circ \mathcal{T}_m$ is also injective and its Jacobian determinant factorizes as*

$$
\det J_{\mathcal{T}} = \det J_{\mathcal{T}_m} \sqrt{\det \left( \left( J_{\mathcal{T}_d} J_{\mathcal{T}_{m \to d}} \right)^T J_{\mathcal{T}_d} J_{\mathcal{T}_{m \to d}} \right)}.
\tag{7}
$$

*Proof.* The injectivity of $\mathcal{T}$ is trivial since it is by definition a composition of injective functions. Since $\mathcal{T}$ is injective, its Jacobian matrix $J_{\mathcal{T}} \in \mathbb{R}^{d \times m}$ is not squared and we cannot use the usual property of bijections in Eq.(1). Instead, we use the definition of Jacobian determinant for injective functions in Eq. (2):

$$
\det J_{\mathcal{T}} = \sqrt{\det \left( J_{\mathcal{T}}^T J_{\mathcal{T}} \right)} = \sqrt{\det \left( \left( J_{\mathcal{T}_d} J_{\mathcal{T}_{m \to d}} J_{\mathcal{T}_m} \right)^T \left( J_{\mathcal{T}_d} J_{\mathcal{T}_{m \to d}} J_{\mathcal{T}_d} \right) \right)},
\tag{8}
$$

where $J_m \in \mathbb{R}^{m \times m}$, $J_{\mathcal{T}_{m \to d}} \in \mathbb{R}^{d \times m}$ and $J_{\mathcal{T}_d} \in \mathbb{R}^{d \times d}$. We now show that we can factor out the Jacobian determinant of $\mathcal{T}_m$, i.e. the bijection that precedes the dimensional inflation step with $\mathcal{T}_{m \to d}$. To do so we use the property that for square matrices $A, B$ $\det(AB) = \det A \det B$ and that $\det A = \det A^T$:

$$
\begin{aligned}
\det J_{\mathcal{T}} &= \sqrt{\det \left( J_{\mathcal{T}_m}^T J_{\mathcal{T}_{m \to d}}^T J_{\mathcal{T}_d}^T J_{\mathcal{T}_d} J_{\mathcal{T}_{m \to d}} J_{\mathcal{T}_m} \right)} \\
&= \sqrt{\det J_{\mathcal{T}_m}^T \det \left( J_{\mathcal{T}_m}^T J_{\mathcal{T}_{m \to d}}^T J_{\mathcal{T}_d}^T J_{\mathcal{T}_d} J_{\mathcal{T}_{m \to d}} \right) \det J_{\mathcal{T}_m}} \\
&= \det J_{\mathcal{T}_m} \sqrt{\det \left( \left( J_{\mathcal{T}_d} J_{\mathcal{T}_{m \to d}} \right)^T \left( J_{\mathcal{T}_d} J_{\mathcal{T}_{m \to d}} \right) \right)} \\
&= \det J_{\mathcal{T}_m} \det J_{\mathcal{T}_d \circ \mathcal{T}_{m \to d}} .
\end{aligned}
\tag{9}
$$

$\blacksquare$

## A.3 AUXILIARY THEOREMS: ADJUGATE MATRIX AND PSEUDO-DETERMINANT

**Theorem 2.** *(Castillo and Zaballa, 2022) Let $A \in \mathbb{R}^{d \times d}$ and let $\lambda \in \mathbb{R}$ be an eigenvalue of A. Let $v, w \in \mathbb{R}^d$ be a right and a left eigenvector, respectively, of A for $\lambda$. Then*

$$w^T v \, \mathrm{adj}(\lambda \mathbb{I}_d - A) = p'_A(\lambda) v w^T \ . \tag{10}$$

*where $p'_A(\lambda)$ is the derivative of the characteristic polynomial $p_A(\lambda) = \det(\lambda \mathbb{I}_d - A)$.*

**Lemma 1.** *Consider the special case where $A \in \mathbb{R}^{d \times d}$ and $\mathrm{rank}\, A = d - 1$ or, in other words, the nullspace of A is one dimensional. Then*

$$\mathrm{adj}(A) = \frac{\det^+(A)}{w^T v} v w^T \ , \tag{11}$$

*where $\det^+$ is the pseudo-determinant.*

*Proof.* Since $\mathrm{rank}\, A = d - 1$, then there exists one zero eigenvalue. For $\lambda = 0$ Theorem 2 reduces to $w^T v \, \mathrm{adj}(-A) = p'_A(0) v w^T$. We can now use the following property of the adjugate matrix: $\mathrm{adj}(cA) = c^d \, \mathrm{adj}(A)$ for any scalar $c$. As a particular case, for $c = -1$ we have $\mathrm{adj}(-A) = (-)^d \, \mathrm{adj}(A)$. Therefore, we obtain that $w^T v \, \mathrm{adj}(A) = (-)^d p'_A(0) v w^T$. Now, the pseudo-determinant is equal to the smallest non-zero coefficient of the characteristic polynomial $p(\lambda) = \det(\lambda \mathbb{I}_d - A)$(Knill, 2014). If we expand the definition we obtain $p(\lambda) = (-)^d p(A - \lambda \mathbb{I}) = p_0 \lambda^d + (-)p_1 \lambda^{d-1} + (-)^k p_k \lambda^{d-k} + (-)^d p_d$ (see Proposition 2, 8. in Knill (2014)). Since $A$ has rank $d - 1$, $p_d = 0$ and the smallest non-zero coefficient is $p_{d-1}$. Finally, note that $p'_A(0) = (-)^d p_{d-1} = (-)^d \det^+(A)$. ∎

**Lemma 2.** *Consider the special case where $A \in \mathbb{R}^{d \times d}$ and $\mathrm{rank}\, A = d - 1$. Then,*

$$\mathrm{Tr}(\mathrm{adj}(A)) = \det^+(A) \ . \tag{12}$$

*Proof.* We take the trace of the left and right-hand side of Eq. (11). We get $\mathrm{Tr}(\mathrm{adj}(A)) = \frac{\det^+(A)}{w^T v} \mathrm{Tr}(v w^T) = \frac{\det^+(A)}{w^T v} \mathrm{Tr}(w^T v) = \det^+(A)$. In the first equality we used the linearity of the trace and factored out the constants $\det^+(A)$ and $w^T v$. Lastly, we used the cyclic property of the trace $\mathrm{Tr}(w^T v) = \mathrm{Tr}(v w^T)$. ∎

## A.4 PROOF OF THEOREM 1

**Theorem 1.** *Let $\mathcal{T} := \mathcal{T}_{\mathrm{s \to c}} \circ \mathcal{T}_r \circ \mathcal{T}_\theta$ as in Figure 2, where $\mathcal{T}_\theta : z \in \mathbb{R}^{d-1} \mapsto \theta \in U_\theta^{d-1}$ is any diffeomorphism to d-spherical angles, $\mathcal{T}_r : \theta \in U_\theta^{d-1} \mapsto [\theta, r(\theta)] \in U_\theta^{d-1} \times \mathbb{R}_{>0}$ with $r(\cdot) : \theta \in U_\theta^{d-1} \mapsto r \in \mathbb{R}_{>0}$ being differentiable, and $\mathcal{T}_{\mathrm{s \to c}} : [\theta, r(\theta)]^T \in U_\theta^{d-1} \times \mathbb{R}_{>0} \mapsto x \in \mathbb{R}^d$ the d-spherical to Cartesian transformation (see Definition 2 in Appendix A.1).*

*Then, the Jacobian determinant of the full transformation $\mathcal{T}$ is equal to*

$$\det\left(J_{\mathcal{T}}\right) = \det\left(J_{\mathcal{T}_\theta}\right) \det\left(J_{\mathcal{T}_{\mathrm{s \to c}}}\right) \|\left(J_{\mathcal{T}_{\mathrm{s \to c}}}^T\right)^{-1} y\|_2 \ , \tag{4}$$

*where $y := \left[-\nabla_\theta r(\theta), 1\right]^T$. Relevantly, $\det J_{\mathcal{T}}$ can be computed exactly and efficiently in $O(d^2)$.*

*Proof.* We start the proof by noting that the transformation $\mathcal{T} := \mathcal{T}_{\mathrm{s \to c}} \circ \mathcal{T}_r \circ \mathcal{T}_\theta$ is injective because $\mathcal{T}_\theta$ and $\mathcal{T}_{\mathrm{s \to c}}$ are bijective and $\mathcal{T}_r$ is injective. The injectivity of $\mathcal{T}_r$ can be easily seen by noting that $\theta \neq \theta' \implies [\theta, r(\theta)] \neq [\theta', r(\theta')]$, independently of $r(\cdot)$. Since the Jacobian matrix $J_{\mathcal{T}} \in \mathbb{R}^{d \times d-1}$ is not squared, we cannot use the usual property of bijections in Eq. (1). Instead, we use the definition of Jacobian determinant for injective functions in Eq. (2):

$$\det J_{\mathcal{T}} = \sqrt{\det\left(J_{\mathcal{T}}^T J_{\mathcal{T}}\right)} = \sqrt{\det\left(\left(J_{\mathcal{T}_{\mathrm{s \to c}}} J_{\mathcal{T}_r} J_{\mathcal{T}_\theta}\right)^T \left(J_{\mathcal{T}_{\mathrm{s \to c}}} J_{\mathcal{T}_r} J_{\mathcal{T}_\theta}\right)\right)} \ , \tag{13}$$

where $J_{\mathcal{T}_r} \in \mathbb{R}^{d \times d-1}$ and $J_{\mathcal{T}_{s \to c}} \in \mathbb{R}^{d \times d}$. According to Remark 1 we can factor out the Jacobian determinant of $\mathcal{T}_\theta$, i.e. the bijection that precedes the dimensional inflation step with $\mathcal{T}_r$:

$$\det J_{\mathcal{T}} = \det J_{\mathcal{T}_\theta} \det J_{\mathcal{T}_{s \to c} \circ \mathcal{T}_r} = \det J_{\mathcal{T}_\theta} \sqrt{\det \left( \left( J_{\mathcal{T}_{s \to c}} J_{\mathcal{T}_r} \right)^T \left( J_{\mathcal{T}_{s \to c}} J_{\mathcal{T}_r} \right) \right)}. \quad (14)$$

The term $\det J_{\mathcal{T}_\theta}$ is the standard Jacobian determinant for bijective layers and can be computed efficiently. We are then left to compute $\det J_{\mathcal{T}_{s \to c} \circ \mathcal{T}_r}$. We now consider the matrix $\tilde{J}_{\mathcal{T}_r} := [J_{\mathcal{T}_r} \; \mathbf{0}_{d \times 1}] \in \mathbb{R}^{d \times d}$ and substitute the determinant with the pseudo-determinant:

$$\det J_{\mathcal{T}_{s \to c} \circ \mathcal{T}_r} = \sqrt{\det \left( J_{\mathcal{T}_r}^T J^* J_{\mathcal{T}_r} \right)} = \sqrt{\det^+ \left( \tilde{J}_{\mathcal{T}_r}^T J^* \tilde{J}_{\mathcal{T}_r} \right)}, \quad (15)$$

where $J^* := J_{\mathcal{T}_{s \to c}}^T J_{\mathcal{T}_{s \to c}} \in \mathbb{R}^{d \times d}$. With $\det^+$ we denote the pseudo-determinant, which is defined as the product of all non-zero eigenvalues. The second equality follows from the fact that $J_{\mathcal{T}_r}^T J^* J_{\mathcal{T}_r}$ and $\tilde{J}_{\mathcal{T}_r}^T J^* \tilde{J}_{\mathcal{T}_r}$ have the same spectrum up to zero eigenvalues, so the determinant of the former coincides with the pseudo-determinant of the latter (by definition). To see that they share the same spectrum up to one zero eigenvalue, consider the explicit structure of the matrix product:

$$\tilde{J}_{\mathcal{T}_r}^T J^* \tilde{J}_{\mathcal{T}_r} = \begin{bmatrix} J_{\mathcal{T}_r}^T J^* J_{\mathcal{T}_r} & \mathbf{0}_{d-1 \times 1} \\ \mathbf{0}_{1 \times d-1} & 0 \end{bmatrix}. \quad (16)$$

The rest of the proof is based on the key observation that $\tilde{J}_{\mathcal{T}_r}$ has rank $d-1$ or, equivalently, that its null space is one-dimensional. As a consequence, we can use Lemma 2 and re-write the pseudo-determinant in terms of the trace of the adjugate matrix:

$$\begin{aligned} \det^+ \left( \tilde{J}_{\mathcal{T}_r}^T J^* \tilde{J}_{\mathcal{T}_r} \right) &= \operatorname{Tr} \left( \operatorname{adj} \left( \tilde{J}_{\mathcal{T}_r}^T J^* \tilde{J}_{\mathcal{T}_r} \right) \right) \\ &= \operatorname{Tr} \left( \operatorname{adj} \left( \tilde{J}_{\mathcal{T}_r}^T \right) \operatorname{adj} \left( J^* \right) \operatorname{adj} \left( \tilde{J}_{\mathcal{T}_r} \right) \right) \\ &= \det \left( J^* \right) \operatorname{Tr} \left( \operatorname{adj} \left( \tilde{J}_{\mathcal{T}_r} \right) \left( J^* \right)^{-1} \operatorname{adj} \left( \tilde{J}_{\mathcal{T}_r}^T \right) \right). \end{aligned} \quad (17)$$

In the second equality we used the property that $\operatorname{adj}(AB) = \operatorname{adj}(B) \operatorname{adj}(A)$ for any $A, B \in \mathbb{R}^{d \times d}$, which easily generalizes to $\operatorname{adj}(ABC) = \operatorname{adj}(C) \operatorname{adj}(B) \operatorname{adj}(A)$. Lastly, if $A$ is invertible, $\operatorname{adj}(A) = \det(A) A^{-1}$. In this case $J^* = J_{\mathcal{T}_{s \to c}}^T J_{\mathcal{T}_{s \to c}}$ has full rank and is thus invertible. Since the trace is a linear operator we can take out $\det(J^*)$, which is a constant.

Since $\tilde{J}_{\mathcal{T}_r}$ has rank $d-1$, its nullspace is one dimensional and we can pick $x \in \mathbb{R}^d \mid \tilde{J}_{\mathcal{T}_r} x = 0$ to span the entire nullspace. The same holds for $\tilde{J}_{\mathcal{T}_r}$, or equivalently for the left nullspace of $J_{\mathcal{T}_r}$, and we can pick $y \in \mathbb{R}^d \mid \tilde{J}_{\mathcal{T}_r}^T y = 0$. We can easily compute $x$ and $y$ by looking at the structure of $\tilde{J}_{\mathcal{T}_r}$:

$$\tilde{J}_{\mathcal{T}_r} = \begin{bmatrix} 1 & 0 & \cdots & 0 & 0 \\ 0 & 1 & & & 0 \\ \vdots & & \ddots & & \vdots \\ 0 & & & 1 & 0 \\ \frac{\partial r}{\partial \theta_1} & \frac{\partial r}{\partial \theta_2} & \cdots & \frac{\partial r}{\partial \theta_{d-1}} & 0 \end{bmatrix} \quad x := \begin{bmatrix} 0 \\ 0 \\ \vdots \\ 0 \\ 1 \end{bmatrix} \quad y := \begin{bmatrix} -\frac{\partial r}{\partial \theta_1} \\ -\frac{\partial r}{\partial \theta_2} \\ \cdots \\ -\frac{\partial r}{\partial \theta_{d-1}} \\ 1 \end{bmatrix}. \quad (18)$$

We now make use of Lemma 1 for $\tilde{J}_{\mathcal{T}_r}$, which gives us

$$\operatorname{adj}(\tilde{J}_{\mathcal{T}_r}) = \frac{\det^+(\tilde{J}_{\mathcal{T}_r})}{y^T x} x y^T. \quad (19)$$

We can now substitute Eq. (19) in Eq. (17):

$$\begin{aligned} \det^+ \left( \tilde{J}_{\mathcal{T}_r}^T J^* \tilde{J}_{\mathcal{T}_r} \right) &= \det \left( J^* \right) \frac{\det^+(\tilde{J}_{\mathcal{T}_r})^2}{(y^T x)^2} \operatorname{Tr} \left( x y^T \left( J^* \right)^{-1} y x^T \right) \\ &= \det \left( J_{\mathcal{T}_{s \to c}}^T J_{\mathcal{T}_{s \to c}} \right) \frac{\det^+(\tilde{J}_{\mathcal{T}_r})^2 x^T x}{(y^T x)^2} \operatorname{Tr} \left( y^T \left( J^* \right)^{-1} y \right) \\ &= \det \left( J_{\mathcal{T}_{s \to c}} \right)^2 \| \left( J_{\mathcal{T}_{s \to c}}^T \right)^{-1} y \|_2^2. \end{aligned} \quad (20)$$

In the first equality we used the fact that $\mathrm{adj}(A^T) = \mathrm{adj}(A)^T$ and we factored out $\mathrm{det}^+(\tilde{J}_{\mathcal{T}_r})^2$ and $(y^T x)^2$, which are constants. In the second equality we used the cyclic property of the trace and factored out $x^T x$. Lastly, we substituted the numerical values $\mathrm{det}^+(\tilde{J}_{\mathcal{T}_r}) = 1$, $y^t x = 1$ and $x^T x = 1$.

We can now analyze the time complexity required to evaluate Eq. (4). The Jacobian determinant for spherical to Cartesian coordinates is known (Muleshkov and Nguyen, 2016)

$$\det J_{s \to c} = (-)^{d-1} r^{d-1} \prod_{k=1}^{d-2} \sin^{d-k-1} \theta_k \tag{21}$$

and can be computed efficiently in $O(d)$ time. Therefore, we only need to show that also $w = \left(J_{\mathcal{T}_{s \to c}}^T\right)^{-1} y$ can be computed efficiently. Solving the full linear system would require a complexity of $O(d^3)$. However, we can exploit the almost-triangular structure of

$$J_{\mathcal{T}_{s \to c}}^T = \begin{bmatrix} \frac{\partial x_1}{\partial \theta_1} & \frac{\partial x_2}{\partial \theta_1} & \cdots & \frac{\partial x_{d-1}}{\partial \theta_1} & \frac{\partial x_d}{\partial \theta_1} \\ 0 & \frac{\partial x_2}{\partial \theta_2} & \cdots & \frac{\partial x_{d-1}}{\partial \theta_2} & \frac{\partial x_d}{\partial \theta_2} \\ \vdots & & \ddots & & \vdots \\ 0 & 0 & & \frac{\partial x_{d-1}}{\partial \theta_{d-1}} & \frac{\partial x_d}{\partial \theta_{d-1}} \\ \frac{\partial x_1}{\partial r} & \frac{\partial x_2}{\partial r} & \cdots & \frac{\partial x_{d-1}}{\partial r} & \frac{\partial x_d}{\partial r} \end{bmatrix} \tag{22}$$

to solve the linear system in $O(d^2)$. One possibility is to perform one step of Gaussian elimination, which requires $O(d^2)$, and make the linear system triangular. The resulting triangular system can be solved in $O(d^2)$. Note that we can compute $J_{\mathcal{T}_{s \to c}}$ very efficiently and analytically (see Eq. (24)), without requiring autograd computations. Overall, the determinant of the full transformation $\mathcal{T}$ can be obtained as

$$\det J_{\mathcal{T}} = \det J_{\mathcal{T}_\theta} \det \left(J_{\mathcal{T}_{s \to c}}\right)^2 \|\left(J_{\mathcal{T}_{s \to c}}^T\right)^{-1} y\|_F^2 \tag{23}$$

and can be computed efficiently in $O(d^2)$. ∎

## A.5 Implementation details

**Implementation of injective flows for star-like manifolds** We provide some details about the implementation of the proposed injective flows and particularly for star-like manifolds in Cartesian coordinates as in Figure 2. We implement the layers in three steps:

- **bijective layers $\mathcal{T}_z$ and $\mathcal{T}_\theta$.** The first bijection $\mathcal{T}_z : z \mapsto z'$ consists of arbitrary (conditional) bijective layers conditioned on the parameter $\lambda$. The conditioning is realized with an expressive Residual network. Then, $\mathcal{T}_\theta : z' \mapsto \theta$ maps the transformed $z'$ into spherical angles $\theta \in U_\theta^{d-1}$. This last transformation is also a bijection that can be implemented with an element-wise non-linear activation like Sigmoid (hence diagonal Jacobian). Otherwise, one could use a base distribution which is already defined on the $d-1$ spherical angles and use a bijective transformation that transforms $\theta$ within their domain $U_\theta^{d-1}$ as $\mathcal{T}_{\mathrm{circ}} : \theta \in U_\theta^{d-1} \mapsto \theta' \in U_\theta^{d-1}$. In short, any bijective layers followed by a element-wise non linear function that maps to spherical angles could be used. We use the circular spline layers proposed by Rezende et al. (2020) because they allow to nicely integrate the boundary conditions arising from the use of spherical coordinates. These layers are based on the neural spline layers proposed in Durkan et al. (2019), which consist of a combination of $K$ segments where each segment is a simple rational-quadratic function. The flexibility of the layers increase with $K$. This way the transformation can be designed such that it is monotonically increasing (hence invertible) and such that it fulfills given boundary conditions. The main difference with Durkan et al. (2019) is that circular splines require periodic boundary conditions in order to enforce continuity of the density at the boundary. This way we can define bijections from $[0, 2\pi]$ to itself. As a consequence, circular splines require the base distribution to be defined on $U_\theta^{d-1}$. In practice, we use the distribution of spherical angles, which results in uniform points on the $d-1$ dimensional sphere, and can be implemented efficiently. We use the implementation of circular layers provided in Stimper et al. (2023). Such construction scales well with the dimensions.

- **injective layer** $\mathcal{T}_r$. The injective step $\mathcal{T}_r : \boldsymbol{\theta} \mapsto [\theta, r(\boldsymbol{\theta})]^T$ only consists in padding the spherical angles with some specified radius function $r(\boldsymbol{\theta})$. The specific expression for the radius function depends on the manifold considered and is detailed in Eq. (25) and Eq. (26) for the $l_p$ (pseudo-) norm ball and for the probabilistic simplex $\mathcal{C}^d$, respectively. In variational inference settings $\mathcal{T}_r$ is not a learnable transformation. In maximum likelihood settings trained on samples, if we assume the samples were generated from a $d-1$ star-like manifold, $r(\boldsymbol{\theta})$ can be implemented with a neural network and made learnable. This would allow to learn the manifold and would provide with a very practical global parametrization.

- **bijective layer** $\mathcal{T}_{s\to c}$: the bijective layer $\mathcal{T}_{s\to c} : [\boldsymbol{\theta}, r(\boldsymbol{\theta})] \mapsto \boldsymbol{x}$ simply implements the spherical to Cartesian transformation in Eq. (6), which is a bijection and can be implemented efficiently. $\mathcal{T}_{s\to c}$ is not a trainable transformation.

For the implementation we rely on the (conditional) normalizing flow library *FlowConductor*[1], which was introduced in Negri et al. (2023) and Arend Torres et al. (2024).

**Efficient implementation of the Jacobian of spherical to Cartesian transformation**   In order to compute the determinant in Eq. (4) we need to compute the Jacobian determinant of the transformation from spherical to Cartesian coordinates $J^T_{\mathcal{T}_{s\to c}}$. By looking at the definition of the coordinate transformation in Eq. (6), we can easily derive the following expression:

$$
J^T_{\mathcal{T}_{s\to c}} = \begin{bmatrix} -rs_1 & rc_1c_2 & \cdots & rc_1s_2\ldots s_{d-2}c_{d-1} & rc_1s_2\ldots s_{d-2}s_{d-1} \\ 0 & -rs_1s_2 & \cdots & rs_1c_2\ldots s_{d-2}c_{d-1} & rs_1c_2\ldots s_{d-2}s_{d-1} \\ 0 & 0 & \ddots & \vdots & \vdots \\ 0 & 0 & \ldots & -rs_1s_2\cdots s_{d-2}s_{d-1} & s_1s_2\ldots s_{d-2}c_{d-1} \\ c_1 & s_1c_2 & \cdots & s_1s_2\ldots s_{d-2}c_{d-1} & s_1s_2\ldots s_{d-2}s_{d-1} \end{bmatrix} \tag{24}
$$

where we used the shorthand $s_i = \sin\theta_i$ and $c_i = \cos\theta_i$. This allows to compute $J^T_{\mathcal{T}_{s\to c}}$ extremely efficiently without requiring to use autograd computations and results in a significant speed up.

**Parametrization of $l_p$ (pseudo-) norm balls**   Here we show how to parametrize the $l_p$ (pseudo-) norm balls in spherical coordinates. Let the $l_p$ (pseudo-) norm of $\boldsymbol{x} \in \mathbb{R}^d$ be defined as $\|\boldsymbol{x}\|_p = (|x_1|^p + \ldots + |x_d|^p)^{1/p}$ with $p > 0$. We consider now the manifold defined by $\|\boldsymbol{x}\|_p = t$ for some $k \in \mathbb{R}_{>0}$. If we write $\boldsymbol{x}$ in spherical coordinates according to Eq. (6), we can take the radius $r$ outside of the norm and express it as a function of the $d-1$ spherical angles as:

$$
r(\theta_1, \ldots, \theta_{d-1}) = \frac{t}{\left( |\cos\theta_1|^p + \sum_{i=2}^{d-1} \left| \cos\theta_i \prod_{k=1}^{i-1} \sin\theta_k \right|^p + \left| \prod_{k=1}^{d-1} \sin\theta_k \right|^p \right)^{1/p}} . \tag{25}
$$

We can use this expression to parametrize the $l_p$ norm balls with the proposed injective flows. Similarly, we can also parametrize the probabilistic simplex $\mathcal{C}^d$. To see this consider the $l_1$ norm ball $\|\boldsymbol{x}\|_1 = |x_1| + \ldots + |x_d|$. If we restrict the domain to the positive quadrant $\boldsymbol{x} \in \mathbb{R}^d_{\geq 0}$ and set the norm to 1, the resulting manifold is defined as $\|\boldsymbol{x}\|_1 = x_1 + \ldots + x_d = 1$ and coincides with $\mathcal{C}^d$. The radius is then parametrized by

$$
r(\theta_1, \ldots, \theta_{d-1}) = \frac{1}{\cos\theta_1 + \sum_{i=2}^{d-1} \cos\theta_i \prod_{k=1}^{i-1} \sin\theta_k + \prod_{k=1}^{d-1} \sin\theta_k} \quad \text{with} \quad \theta_i \in [0, \pi/2] \,\forall i , \tag{26}
$$

where the constraint on the angles enforces $\boldsymbol{x} \in \mathbb{R}^d_{\geq 0}$. Note that it is straightforward to analytically derive the expression for the partial derivatives $\frac{\partial r}{\partial \theta_i}$ in Eq. (26). This makes the computation of $y$ in Eq.(4) more efficient than computing the gradients with autograd and results in a speed up.

---

[1] https://github.com/FabricioArendTorres/FlowConductor

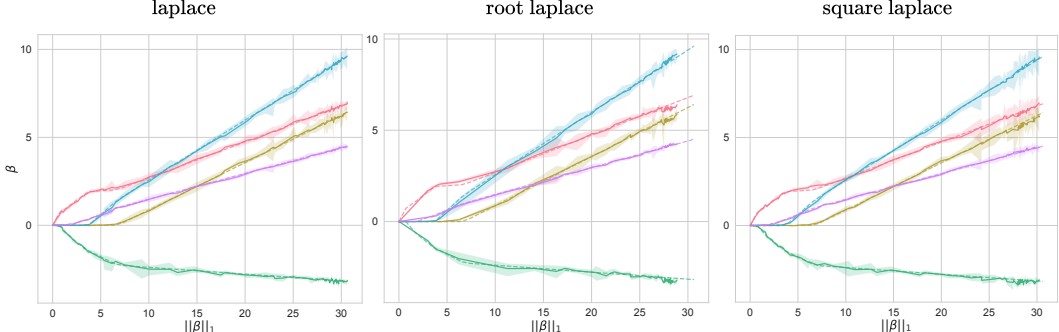

Figure 8: MAP solution for Laplace prior, root laplace and square laplace, which are all monotonic transformation of the Laplace distribution. The MAP solution paths coincide.

## A.6 APPLICATIONS: FURTHER DETAILS

### A.6.1 ARCHITECTURE

We use two different architectures. One is for standard NFs that we use for the subjective penalized likelihood regression problem. The other architecture is the injective flow that is used for the objective Bayes version of the regression problem and the portfolio diversification application.

**Standard NF**  It consists of a normal distribution as base distribution. Then we use 5 blocks of permutation transformation, a sum of Sigmoids layer (Negri et al., 2023) and an activation norm. The sum of Sigmoid layer consists each of 30 individual Sigmoid functions in three blocks.

**Injective flows**  The base distribution is either the probabilistic simplex or the complete $\|\boldsymbol{\beta}\|_1 = 1$ depending on the application. We follow this with again 5 layers of the circular bijective layers (Rezende et al., 2020), each consisting of three blocks with 8 bins. At the end these values are mapped to Cartesian coordinates with the proposed dimensionality inflation step.

### A.6.2 TRAINING

Both the standard NFs and the injective flows are trained by minimizing the reverse KL divergence with respect to the (unnormalized) target density $p(\boldsymbol{x})$:

$$q_{\theta^*}(\boldsymbol{x}) = \arg\min_{\theta \in \Theta} \mathrm{KL}\big(q_\theta(\boldsymbol{x})||p(\boldsymbol{x})\big) = \arg\min_{\theta \in \Theta} \mathbb{E}_{\boldsymbol{x} \sim q_\theta}\left[\log \frac{q_\theta(\boldsymbol{x})}{p(\boldsymbol{x})}\right]. \tag{27}$$

We optimize the reverse KL divergence using Adam (Kingma and Ba, 2017) as optimizer with default parameters. Notably, all trained flows converged in a matter of minutes on a standard consumer-grade GPU (RTX2080Ti in our specific case).

### A.6.3 PENALIZED LIKELIHOOD REGRESSION

In the next paragraph we provide further details on the experiment introduced in Section 5.3, which involves the penalized likelihood model defined in Eq. (5).

**Synthetic dataset creation**  The synthetic regression dataset is created by sampling $X^*$ from a 5 dimensional Wishart distribution $W_5(7, I)$. The response variable $y$ is then created by $X^*\boldsymbol{\beta}^* + \epsilon$ where $\boldsymbol{\beta}^*$ is standard normal distributed and $\epsilon$ is normal distributed with zero mean and a standard deviation of 4.0.

**Subjective Bayes**  The subjective Bayes relies on a prior $p(\boldsymbol{\beta}|\lambda)$. The Laplace prior is given by

$$p_{lap}(\boldsymbol{\beta}|\lambda) \propto \prod_i \exp\{-\lambda|\beta_i|^p\}. \tag{28}$$

The two other test priors are $p_{sq}(\boldsymbol{\beta}|\lambda) \propto p_{lap}(\boldsymbol{\beta}|\lambda)^2$ and $p_{rt}(\boldsymbol{\beta}|\lambda) \propto p_{lap}(\boldsymbol{\beta}|\lambda)^{1/2}$. Any monotonic transformation may change the $\lambda$-axis but leave the MAP solution path unchanged. This can be

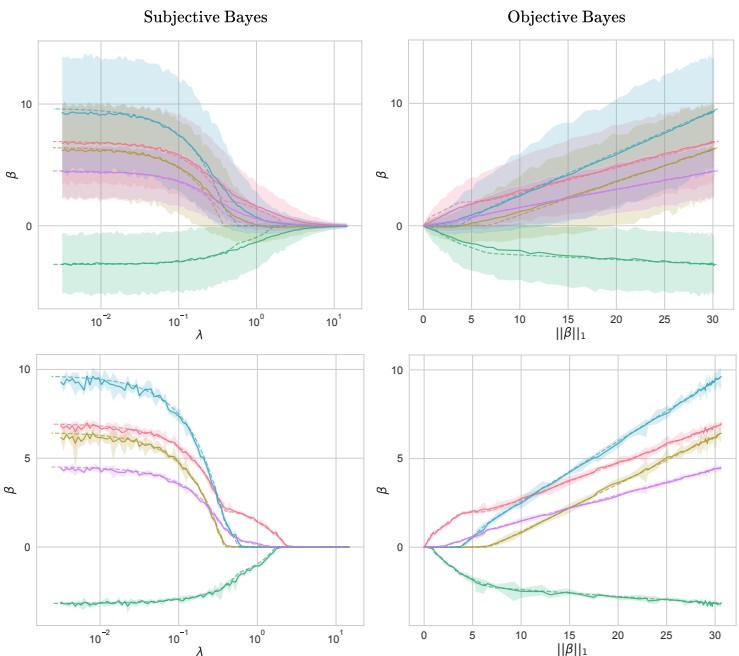

Figure 9: Solution paths for subjective prior as a function of $\lambda$, and objective priors, as a function of the norm $\|\boldsymbol{\beta}\|_1$. Below we report the solution paths in the MAP limit.

seen in Figure 8 where we show the MAP solution path for different subjective priors. For this visualization we reparametrize the axis such that the $\lambda$-axis is transformed into a $\|\boldsymbol{\beta}\|_1$-axis. This makes clear that the solution paths are equivalent.

**Objective Bayes** The objective Bayes approach circumvents the definition of $p(\boldsymbol{\beta}|\lambda)$. The flow is directly defined on the manifolds coinciding with the contour lines of $p(\boldsymbol{\beta}|\lambda)$. As such, samples from the posterior all share a chosen norm value $\|\boldsymbol{\beta}\|_1 = k$. Figure 9 highlights the different parametrizations of the subjective and objective approach.

### A.7 APPROXIMATE VS EXACT JACOBIAN: SAMPLE QUALITY AND RUNTIME

#### A.7.1 BAYESIAN MIXING MODEL

**Comparison with MCMC sampler** We compare the proposed injective flows with an ad-hoc MCMC sampler in the conjugate case (Dirichlet prior and multinomial likelihood). In this setting we know the posterior analytically and we can compare it to both the MCMC sampler and the proposed injective flow. In this experiment the injective flow is trained to minimize the reverse KL divergence with the unnormalized target posterior (prior times likelihood). We compare the methods for increasing dimensionality ($d = 15, 30, 50$) and in Figure 11 we show $95\%$ credibility intervals of the posterior. Results show that (i) the proposed injective flow correctly estimates the posterior distribution for all dimensions and (ii) it outperforms the MCMC sampler already in low dimensions ($d = 30, 50$). Interestingly, this experiment also shows that our approach is able to learn very sparse solutions characterized by multiple modes.

**Details about the MCMC sampler** In Bayesian mixing models the posterior lives on the simplex, which is known to be challenging for MCMC samplers (Altmann et al., 2014; Baker et al., 2018) because of the constraint that the sampled vector must be positive and sum up to one. Designing proposals close to the boundary is particularly difficult (Baker et al., 2018). For sparse distributions, where mass is concentrated at the boundaries, this is especially noticeable and explains why the MCMC sampler struggles in high dimensions for sparse solutions. What makes a fair comparison difficult is that various existing samplers (Altmann et al., 2014; Baker et al., 2018) are suited for

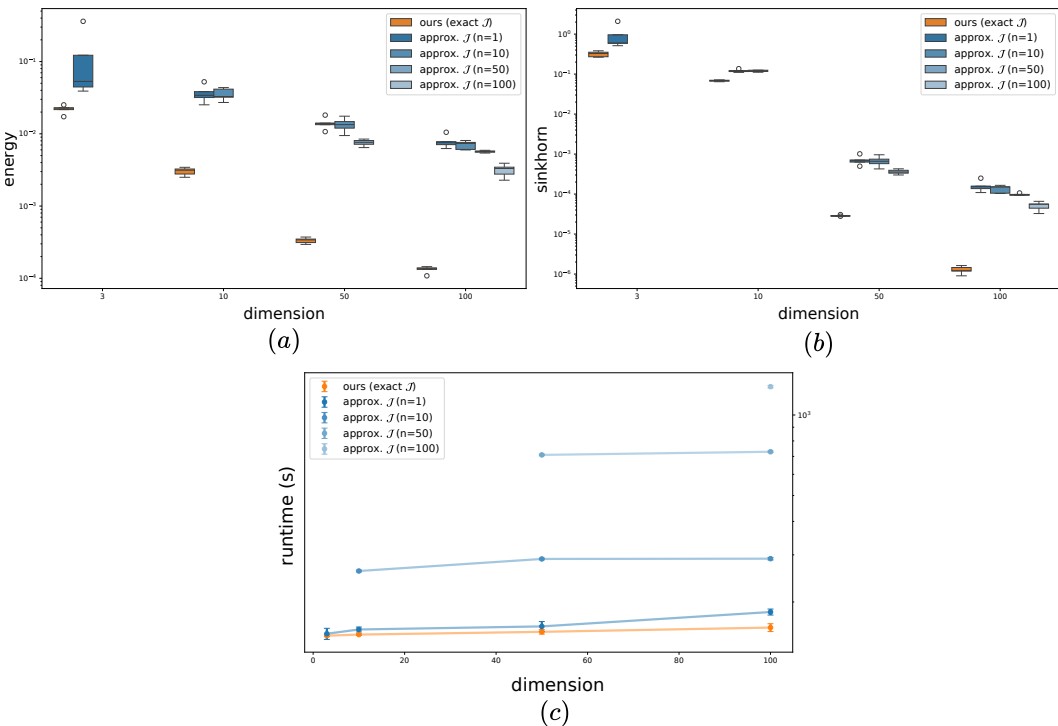

Figure 10: Additional comparison between two identical models trained with the exact Jacobian, as we propose, according to Eq. (4) and with the Hutchinson trace estimator, commonly used in injective papers. The experimental setup is described in Section 5.1. We compare samples of the two models with samples from the true distribution as a function of the dimension and of the number of samples used in the Hutchinson trace estimator ($n = 1, 10, 50, 100$). Note that the number of samples is upper bounded by the dimensionality of the problem. We quantify the similarity of the samples with the samples from the true distribution with (a) the Energy distance MMD ("energy") and (b) the Sinkhorn divergence ("sinkhorn"). The Energy distance MMD is computed using the kernel $-\|x - y\|_2$. The Sinkhorn divergence interpolates between Wasserstein (blur=0) and kernel (blur=$\infty$) distances and we used the default value blur=0.05. Below (c) we show the runtime of our exact model compared to the approximate one as a function of increasing number of samples.

specific families of target distributions. For ideal performance we would need to change the entire sampler depending on the target, while we can use our flow without any fine-tuning.

We compare to a Metropolis Hastings sampler that can handle mixtures of Gaussians targets well in a non-sparse setting. It uses proposals that are by construction on the manifold by sampling from a down-scaled Dirichlet distribution centered on the current state. Despite trying different proposals we did not observe any relevant improvement. As shown, this sampler however struggles when significant sparseness is present. Our sampler uses 1000 chains each with 100'000 samples. In contrast, the proposed injective flow did not require any fine-tuning and learnt sparse solutions. For a fair comparison, we set the runtime of the flow to match that of the MCMC samplers and used roughly similar memory footprint ($\pm 20\%$). Lastly, we checked that running the sampler for a significantly longer time ($5\times$) did not achieve the performance of the flow.

**Portfolio optimization** In portfolio optimization the cumulative return is often of interest. Figure 12 shows the effect of the different priors on the cumulative return. The sparser priors lead to a slightly wider distribution of the return. In this example, this leads to the target index being a closely matched by some of the posterior samples, where the samples of the uniform prior seem to be further away from the target index in some parts of the time interval. The bottom row of the Figure 12 further shows the sampled sparsity patterns. These show that the sparse priors can lead to significantly different mixtures with similar data fitting quality.

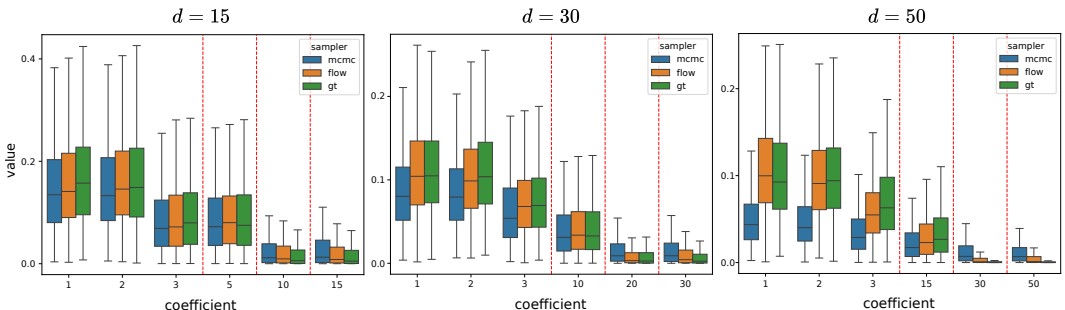

Figure 11: 95% posterior credibility intervals for a Bayesian model with multinomial likelihood and Dirichlet prior. We compare the posterior obtained with samples from an ad-hoc MCMC sampler ("mcmc"), samples from the approximate posterior modeled by the proposed injective flow ("flow") and with the analytical posterior ("gt"). We evaluate the samplers for increasing dimensionality of the problem $d = 15, 30, 50$.

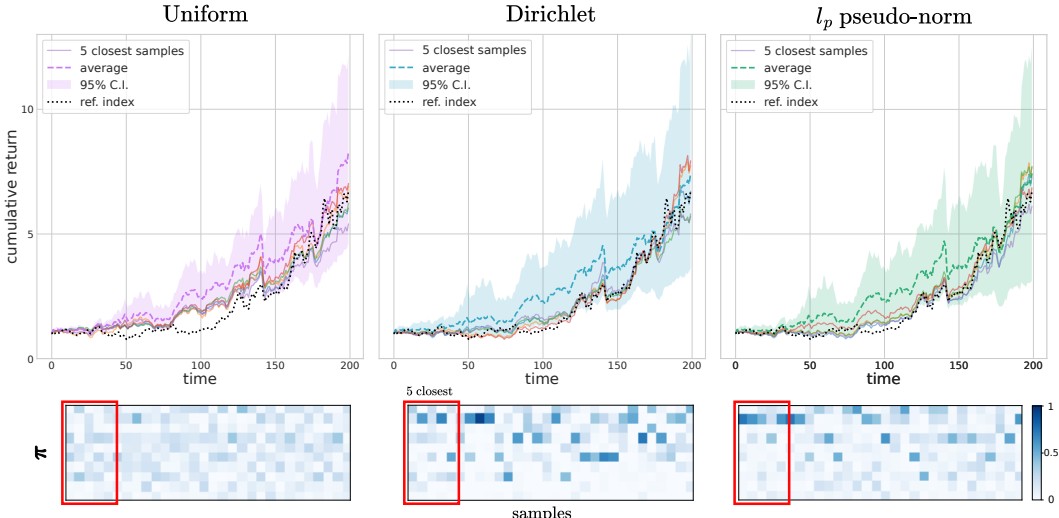

Figure 12: Cumulative return as a function of time. We plot 95% posterior C.I. with the uniform prior, Dirichlet prior and $l_p$-norm prior. We also plot the 5 samples that are the closest to the ground truth cumulative return. In the bottom panel we visualize the weight samples as a heatmap to highlight sparsity patterns.

