# OpenReview forum: "Injective flows for star-like manifolds"
_ICLR.cc/2025/Conference — ICLR 2025 Poster_

### Official Review · Reviewer_xKG1 · 2024-10-17

**Soundness:** 2
**Presentation:** 3
**Contribution:** 3
**Rating:** 6
**Confidence:** 4

**Summary:**

The authors propose an embedding of arbitrary d-1 dimensional star-like manifolds (i.e., deformations of the unit sphere) into the $\mathbb{R}^d$ which allows the computationally efficient evaluation of its determinant. They apply this embedding for constructing normalizing flows on such manifolds, by using an injective mapping from a d-1 dimensional latent space to the manifold. The proposed model is applied to several examples.

**Strengths:**

The paper is easily readable and written in a clear way. As far as I know the approach is new, implementable and computationally tractable. In contrast to prior papers it covers a whole class of manifolds.

**Weaknesses:**

My main critic on the paper is the missing comparison to existing injective flows. The authors emphasize at several points of the paper that it is important to have density evaluations of the model which are i) exact and ii) have improved complexity. As outlined in the literature part of the authors, there exist plenty of methods meeting one of the properties i) or ii). To validate the importance of meeting both at the same time, I would suggest two experiments:

i) Run an experiments to fit a not too simple distribution on a star-like manifold (maybe something like Fig 3b) and train an injective flow once with naive O(d^3) evaluation of the change-of-variables formula and once with the formula proposed in Thm 1. Then, create a plot of execution time vs dimension.

ii) Run an experiment in a similar setup for variational inference. Train an injective flow with exact determinant (as proposed) and some approximate determinant (e.g., one of the models which are cited in the related-work part) and compare the quality of generated samples (e.g. in Wasserstein/MMD or a similar two-sample comparison of distributions). Following the authors claims, the injective flow with exact determinant should lead to more accurate results than the approximate.

Even though the existing numerical experiments provide some proof-of-concept, I think that the paper could benefit a lot by evaluating more the proposed architecture itself in terms of speed and accuracy instead of using it for specific applications. The comparison with the completely different MCMC samplers looks a bit out of place since it is a completely different model which is known not to scale well along the dimensions (which is also observed in Fig 9).

Additionally, I would like to point the authors to the papers [1,2]. [1] considers a specific architectures which makes the evaluation of the necessary determinant cheap and easy, [2] which trains the normalizing flows with a VAE loss.

From a technical side, I am bit confused since the change from spherical to Cartesian coordinates is not a bijection and definitely not a diffeomorphism. The vectors $(0,\theta_2,...,\theta_{d-1},1)$ map all to the same point in the $\mathbb{R}^d$ regardless of the choice of $\theta_2,...,\theta_{d-1}$. Certainly, one can exclude sets of measure zero from domain and image of this transformation such that it becomes bijective, but then still the determinant of this transform becomes arbitrary close to zero when $\theta_1$ approaches zero such that the inverse is no longer a $C^1$-function (and hence the embedding not a $C^1$ diffeomorphism). I am not completely sure if this can cause limitations in actual applications. But the issue should be treated more thoroughly in the paper. In fact, it is easy to see that there exists no diffeomporphism from $\mathbb{R}^{d-1}$ to a sphere (or any other star-like manifold) since those have different topologies (e.g., $\mathbb{R}^{d-1}$ is contractible in contrast to star-like manifolds). A possible way to overcome such issue is the use of a mixture of several embeddings (for star-like manifolds two should be enough), see [3]. As the authors do not need too many geometrical properties of their embedding an "we exclude zero sets"-solution might be enough, but again, this issue should be discussed in the paper.

In summary, I think some numerical comparisons to existing works are necessary such that I think that the paper is below the acceptance threshhold, but I am willing to raise my score if the critics from above is addressed.


[1] Kothari et al. "Trumpets: Injective Flows for Inference and Inverse Problems", UAI 2021

[2] Chen et al. Learning flat latent manifolds with VAEs, ICML 2020

[3] Alberti et al. "Manifold Learning by Mixture Models of VAEs for Inverse Problems", JMLR 2024

**Questions:**

I outlined my questions and suggestions in the "weaknesses" part.

---

> ### Author Response · Authors · 2024-11-19
>
> We thank the reviewer for the very insightful comments and for suggesting in detail which experiments would strengthen the present work, which we implemented.
>
> ## Weaknesses:
> > The authors emphasize at several points of the paper that it is important to have density evaluations of the model which are i) exact and ii) have improved complexity. [...] I would suggest two experiments:
>
> > 1. Run an experiments to fit a not too simple distribution on a star-like manifold (maybe something like Fig 3b) and train an injective flow once with naive O(d^3) evaluation of the change-of-variables formula and once with the formula proposed in Thm 1. Then, create a plot of execution time vs dimension.
>
> We thank the reviewer for suggesting an insightful experiment. We agree that an empirical runtime comparison adds significance to the proposed method. We followed the suggestions in detail and computed the Jacobian determinant explicitly (Eq (2)) and with the proposed method (Theorem 1 Eq (4)). The empirical results confirm the claim in Theorem 1 and show a significant speedup of the proposed approach.
>
> >2. Run an experiment in a similar setup for variational inference. Train an injective flow with exact determinant (as proposed) and some approximate determinant (e.g., one of the models which are cited in the related-work part) [...]. Following the authors claims, the injective flow with exact determinant should lead to more accurate results than the approximate.
>
> We thank the reviewer for suggesting another insightful comparison. We followed the suggestions and compared the MSE of the learned density when we use the proposed approach (Theorem 1) or the Hutchinson trace estimator, which is the most common approximation in injective flow papers [3,4,5], including state-of-the-art work [5]. Our results show that already for trivial densities (uniform) on a simple manifold (lp-pseudo norm ball with $p=0.5$) training with the approximate Jacobian leads to significantly worse results. The difference in MSE with our exact approach increases even further as the dimension increases.
>
> > Even though the existing numerical experiments provide some proof-of-concept, I think that the paper could benefit a lot by evaluating more the proposed architecture itself in terms of speed and accuracy instead of using it for specific applications. The comparison with the completely different MCMC samplers looks a bit out of place since it is a completely different model which is known not to scale well along the dimensions (which is also observed in Fig 9).
>
> We have now provided some additional experiments that show why the exact Jacobian is useful and that we can actually compute it much more efficiently than with the explicit computation. As the reviewer pointed out, this helps highlight the relevance of the proposed approach.
>
> The comparison with the MCMC sampler was not meant for this purpose but rather to show that the proposed approach is not only competitive with other standard methods like MCMC, but that it better scales with dimensionality.
>
> > Additionally, I would like to point the authors to the papers [1,2]. [1] considers a specific architecture, which makes the evaluation of the necessary determinant cheap and easy, [2] which trains the normalizing flows with a VAE loss.
>
> We thank the reviewer for highlighting relevant related work. In [1] the authors still use an approximation of the Jacobian determinant and employ a bound obtained by approximating the Jacobian determinant as in standard normalizing flows: as the product of the Jacobian of individual layers. It is not surprising that this works well mainly for MAP estimates, which are however restrictive. We added reference [1] to our paper. In [2] a bound on the log-likelihood method is used. The evaluation of the Jacobian is however not cheap (since they still need to explicitly compute the metric tensor $J^TJ$), so they resort to a first order approximation.
>
> > From a technical side, I am bit confused since the change from spherical to Cartesian coordinates is not a bijection and definitely not a diffeomorphism. [...].
>
> We agree with the reviewer that, technically speaking, the transformation from spherical to Cartesian coordinates is not a bijection. For instance, Cartesian coordinates at the poles do not map to unique angles. However, this is a zero-measure set, which we can very easily avoid in our implementation by offsetting with a small epsilon. We agree with the reviewer that this aspect should be mentioned in the paper, so we included this discussion in Section 3.4.
>
> [1] Kothari et al. "Trumpets: Injective Flows for Inference and Inverse Problems", UAI 2021
>
> [2] Chen et al. Learning flat latent manifolds with VAEs, ICML 2020
>
> [3] Caterini et al, Rectangular flows for manifold learning NeurIPS, 2021
>
> [4] Flouris et al., Canonical normalizing flows for manifold learning, NeurIPS, 2023
>
> [5] Sorrenson et al., Lifting architectural constraints of injective flows, ICLR, 2024

---

> ### Comment · Reviewer_xKG1 · 2024-11-21
> **Response to the Authors**
>
> Thank you for the additional explanations. I think the new version goes definitely in the right direction. However, I have a couple of follow-up questions and comments:
>
> I think the experimental setup of Fig 3 (particularly part b) requires more explanations and details (which of course can be postponed to the appendix, if required for the page limit). In particular:
>
> - which architecture are you using (hopefully the same for both training methods)?
>
> - How do you implement the Hutchinson trace estimator (Rademacher or Gauss vectors, how many random vectors)
>
> - Do you use more random vectors in Hutchinson for evaluation than for training? (this certainly should be done)
>
> - Can you see this quality difference also for samples generated from these flows or is it an issue of evaluating the density of the arising model (so comes the larger error for the inexact logdet from the training or the evaluation phase)? You could plot generated samples in 2- and 3-D (which you should do anyway) and evaluate some metric like MMD between samples generated from the flows and the ground truth.
>
> - What does "fails to converge" mean? The Hutchinson estimator does not converge (would be weird)? Or the training loss does not converge (would this really be an issue? the Hutchinson estimator is still unbiased, even though it certainly increases the variance of the loss)?
>
> - Why do you use such a simple target density? I do not think that this has to be something very complicated, but the uniform distribution is that simple that I wonder if there is a reason behind that.
>
> - In Fig 3a): Why is the naive computation complexity $O(d^{3.4})$ and the complexity of your method $O(d^{1.5})$? Before I found the values $O(d^3)$ and $O(d^2)$ in the paper. Are these value determined numerically? If this is the case make this clear (particularly since it is quite a large variation between theoretical complexity and implementation).
>
> I am also not really satisfied with the treatment of the change from spherical to Cartesian coordinates. As I have already written, I do not think that this is an issue from a numerical viewpoint. However, if I would read the paper again from scratch, I would be stumbling at exactly the same point with exactly the same question as for the first time (the intro lets me expect injective transforms and then it comes to something which is obviously not injective without explanation at this point). I would expect the authors to clarify this issue exactly where it appears (at least by pointing to the discussion). In addition, this issue can also be a limitation, whenever some operations are performed in the latent space (which often appears in fine-tuning methods of normalizing flows). In this case, introducing artificial singularities can be limiting.

---

> > ### Author Response · Authors · 2024-11-23
> >
> > We sincerely thank the reviewer for continuing the very nice discussion and we apologize for not including enough details in the rebuttal. Our intention was to summarize the additional experiments within the global response and within the main text. However, we are more than happy to provide further details here and in the appendix. To highlight differences with the previous revision, we color-coded the new changes in red.
> >
> > In order to better address the questions raised, we included additional results where we use the approximate Jacobian determinant with $n=1,10,50,100$ Gaussian samples. We also evaluated the performance in terms of the quality of the samples compared to samples from the true distributions. All new results show that our proposed approach provides better solutions in terms of reconstructed density (Figure 3b) and generated samples (Figure 10a and 10b). Runtime comparison confirms the intuition that the approximation works better as the number of samples is increased (Figure 3b, 10a, 10b) but at the cost of significantly increased runtime (Figure 10c).
> >
> > > I think the experimental setup of Fig 3 (particularly part b) requires more explanations and details (which of course can be postponed to the appendix, if required for the page limit). In particular: which architecture are you using (hopefully the same for both training methods)?
> >
> > Yes, we use the same architecture for both models. We use 10 layers of circular splines, each with 5 autoregressive blocks and 10 bins (which is more than sufficient for such a simple settings). The only thing that changes between the two models is that during training in one case we compute the Jacobian determinant as in Eq. (4) (Theorem 1) and in the other we approximate its gradients with the Hutchinson trace estimator. In particular, we use the state-of-the-art approach in [1,2,3] (and their code as well https://github.com/vislearn/FFF).
> >
> > > How do you implement the Hutchinson trace estimator (Rademacher or Gauss vectors, how many random vectors)
> >
> > We use the implementation in [1,2,3] and employ Gauss vectors. We initially used one samples, as suggested by authors in [1,2,3], but we now included a comparison with $n=1,10,50,100$. We observe the expected trend: the approximation leads to an improved density reconstruction and to more accurate samples. Note that, as the number of samples is increased, the runtime is not anymore comparable with our method (Figure 10c). So even if the approximation leads to a better log likelihood, which is expected (but still worse than ours), this comes at a much higher computational cost. This makes our approach much more interesting and viable in practice.
> >
> > > Do you use more random vectors in Hutchinson for evaluation than for training? (this certainly should be done)
> >
> > For evaluation we use the exact Jacobian determinant also for the approximate method, since at that point the computational cost is not a bottleneck anymore.
> >
> > > Can you see this quality difference also for samples generated from these flows or is it an issue of evaluating the density of the arising model (so comes the larger error for the inexact logdet from the training or the evaluation phase)? You could plot generated samples in 2- and 3-D (which you should do anyway) and evaluate some metric like MMD between samples generated from the flows and the ground truth.
> >
> > Yes, the quality difference is also reflected in the samples. As suggested, we now measured the quality of samples and compared to the samples generated from the
> > true distribution. We used two metrics: the energy distance MMD, computed using the
> > kernel $−∥x − y∥_2$, and the Sinkhorn divergence, which interpolates between Wasserstein (blur=0) and kernel
> > (blur=∞) distances (we used the default value blur=0.05). Both metrics show that the quality of samples generated with our method is superior than with the approximate Jacobian determinant, even when the number of samples in the estimator is increased. The trend shows that the quality increases with the number of samples, but at the cost of increased runtime.
> >
> > > What does "fails to converge" mean? The Hutchinson estimator does not converge (would be weird)? Or the training loss does not converge (would this really be an issue? the Hutchinson estimator is still unbiased, even though it certainly increases the variance of the loss)?
> >
> > We apologize for the confusing expression. With "fails to converge" we meant that the loss did not converge. We further investigate the problem and trained the approximate model on a larger GPU and with a smaller learning rate. This way we could solve the convergence issue we faced in higher dimensions. We believe the problem was caused by the fact that because of the variance of the estimator it is preferable to take smaller gradient steps. Finally, we note that our model didn't require any fine-tuning and worked immediately.

---

> > > ### Author Response · Authors · 2024-11-23
> > >
> > > > Why do you use such a simple target density? I do not think that this has to be something very complicated, but the uniform distribution is that simple that I wonder if there is a reason behind that.
> > >
> > > We decided to use the uniform distribution because in this case we can evaluate the log likelihood exactly and because we can generate samples from the true distribution [4]. This way we could evaluate the log density but also the quality of the samples.
> > >
> > > > In Fig 3a): Why is the naive computation complexity $O(d^{3.4})$ and the complexity of your method $O(d^{1.5})$? Before I found the values $O(d^3)$ and $O(d^2)$ in the paper. Are these value determined numerically? If this is the case make this clear (particularly since it is quite a large variation between theoretical complexity and implementation).
> > >
> > > Yes, the theoretical complexities are $O(d^3)$ for the explicit computation and $O(d^2)$, as in Theorem 1. By extrapolating the trend from the log-log plot between the dimensions [7000-12000] we obtained the coefficients 3.4 and 1.5. We now repeated the experiments on a larger GPU that allowed to extend the computation to dimension 25'000. By fitting a linear curve to the log-log plot in the range [15000, 25000] we improved the estimate (Figure 3a). We empirically found that the exact explicit method in Eq. (2) requires $O(d^{2.96})$ and ours in Eq. (2) requires $O(d^{1.81})$.
> > >
> > > > I am also not really satisfied with the treatment of the change from spherical to Cartesian coordinates. As I have already written, I do not think that this is an issue from a numerical viewpoint. However, if I would read the paper again from scratch, I would be stumbling at exactly the same point with exactly the same question as for the first time (the intro lets me expect injective transforms and then it comes to something which is obviously not injective without explanation at this point). I would expect the authors to clarify this issue exactly where it appears (at least by pointing to the discussion). In addition, this issue can also be a limitation, whenever some operations are performed in the latent space (which often appears in fine-tuning methods of normalizing flows). In this case, introducing artificial singularities can be limiting.
> > >
> > > We thank the reviewer for pointing out that we should have the discussion early in the paper and not just as an implementation detail. We now updated the main text accordingly. The hypothesis for the change of variable formula can be relaxed to diffeomorphisms up to zero-measure sets [5] (using Sard's theorem). We highlight this point in the background Section 2. Then, in the proposed method Section 3.2 we note that the spherical to Cartesian coordinate transformation is a diffeomorphism almost everywhere, and hence does not create a problem from a theory stand-point. In other words, the probability computed after the change of variable is still exact since the contribution coming from non-differentiable parts has zero volume. When training for variational inference settings, this means that sampling a singular point is mathematically impossible. Then, in Section 3.3 we argue that in practice we can easily avoid potential numerical instabilities arising from sampling points close to the singularities by offsetting with a small epsilon. As pointed out by the reviewer, the strategy to avoid singularities might create problems in other specific applications. However, other strategies (like clamping) would also be possible depending on the settings.
> > >
> > > We hope that we could clarify the points raised and we would be very happy to continue the discussion further.
> > >
> > > [1] Draxler et al., Free-form flows: Make any architecture a normalizing flow, AISTATS, 2024
> > >
> > > [2] Sorrenson et al, Lifting architectural constraints of injective flows, ICLR, 2024
> > >
> > > [3] Sorrenson et al, Learning Distributions on Manifolds with Free-form Flows, NeurIPS 2024
> > >
> > > [4] Song et al.,Lp-Norm Uniform Distribution, Proceedings of the American Mathematical Society, 1997
> > >
> > > [5] Spivak, Calculus on Manifolds, Westview Press, 1965

---

> ### Comment · Reviewer_xKG1 · 2024-11-23
>
> Thanks for the clarifications. I have updated the score.
>
> Nevertheless, I did not expect such a large difference, particularly in the low dimensional setting. I experimented a bit with Hutchinson estimators in the past and, for nearly as many vectors as dimensions, it should be close to exact... Additionally, as a side note: As far as I know, the estimator has a lower variance with Rademacher vectors, which therefore usually converges faster (but the difference should be moderate).

---

> > ### Author Response · Authors · 2024-12-02
> >
> > Thanks for the continued and very insightful discussion! We further investigated and run experiments in the same settings with Rademacher vectors. We obtained very consistent results with what was shown with Gaussian samples as well. The reason is that we use orthogonalized Gaussian noise because in [1] the authors shows it reduces the variance of the estimator. We believe that the trace estimator is not particularly suited for our model because of the particular type of transformation that defines the radius ($T_{r}$) and then converts spherical to Cartesian coordinates ($T_{s \rightarrow c}$).
> >
> > We thank again the reviewer for the engaging discussion that significantly improved our paper presentation and results.
> >
> > [1] Sorrenson et al, Lifting Architectural Constraints of Injective Flows, ICLR 2024

---

### Official Review · Reviewer_xkKJ · 2024-10-28

**Soundness:** 3
**Presentation:** 2
**Contribution:** 2
**Rating:** 6
**Confidence:** 3

**Summary:**

This paper addresses the substantial computational cost associated with normalizing flows when applied to densities on manifolds. The proposed method is specifically tailored for star-like manifolds and efficiently computes the Jacobian determinant, outperforming standard normalizing flows. Furthermore, the paper introduces Bayesian methods for penalized likelihood models, where penalty level-sets are regarded as start-like manifolds, and variational inference methods for probabilistic mixture models.

**Strengths:**

- The proposed injective flow method for star-like manifolds significantly reduces the standard complexity of $O(d^3)$ to $O(d^2)$.
- The injective flow approach appears to be both intriguing and novel, and the proof presented is sound.
- The paper outlines several potential applications, including the objective Bayesian approach and posterior inference in probabilistic mixing models.

**Weaknesses:**

- Experiments are relatively limited, and no baselines are employed for comparison apart from the MCMC sampler in FIgure 9. I would suggest state-of-the-art normalizing flow methods and possibly other generative models for the baselines.
- The purported efficiency enhancement from $O(d^3)$ to $O(d^2)$ is not substantiated by experimental results. The authors could include runtime comparisons between their method and standard normalizing flows for increasing dimensionality $d$. This would provide empirical evidence for the claimed computational improvement.
- A minor issue: the caption of Figure 6 should include the correct probability distribution: $p(\boldsymbol\pi)|\mathcal{D}$.

**Questions:**

- In Theorem 1, it is stated that $\mathcal T_\theta$ can be an arbitrary diffeomorphism in $\mathbb{R}^{d-1}$. However, to the best of my understanding of normalizing flows, the evaluation of its Jacobian could be as computationally expensive as $O(d^2)$ if not specifically designed. Do you require any constraints on $\mathcal{T}_\theta$?
- How do the properties of the manifold influence the training process of the flow model?

---

> ### Author Response · Authors · 2024-11-19
>
> We thank the reviewer for suggesting further experiments to strengthen our approach, which we implemented, and for highlighting the importance of an empirical runtime comparison.
>
> ## Weaknesses:
> > Experiments are relatively limited, and no baselines are employed for comparison apart from the MCMC sampler in Figure 9. I would suggest state-of-the-art normalizing flow methods and possibly other generative models for the baselines.
>
> We thank the reviewer for suggesting further experiments to validate the proposed approach. While most of the work on injective flows focuses on maximum likelihood settings (as other generative models), we could still apply the most common approximation of the Jacobian determinant to variational inference as well. In particular, we used the Hutchinson trace estimator employed in many injective flow papers [1,2,3], including state-of-the-art flows [3]. In Figure 3b we show that such approximation during training leads to inaccurate density estimation also at test time (when the exact Jacobian can be used). The performance degrades further if the dimensionality increases. In the general response we argue in more details, we do not focus on maximum likelihood setting but rather on variational inference.
>
> > The purported efficiency enhancement from $O(d^3)$ to $O(d^2)$ is not substantiated by experimental results. The authors could include runtime comparisons between their method and standard normalizing flows for increasing dimensionality $d$. This would provide empirical evidence for the claimed computational improvement.
>
> We thank the reviewer for highlighting that the claimed result in Theorem 1 should be validated experimentally. As suggested, we compared the proposed approach to the explicit computation of the exact Jacobian (Eq (2)) and we showed empirically that we indeed significantly improve the complexity from cubic to quadratic.
>
>
> > A minor issue: the caption of Figure 6 should include the correct probability distribution: $p(\pi|\mathcal{D})$
>
> We thank the reviewer for noticing the typo. We have updated the pdf accordingly.
>
> ## Questions:
>
> > In Theorem 1, it is stated that $T_\theta$ can be an arbitrary diffeomorphism in $\mathbb{R}^{d-1}$. However, to the best of my understanding of normalizing flows, the evaluation of its Jacobian could be as computationally expensive as $O(d^2)$ if not specifically designed.
>
> Yes, standard normalizing flows can be as expensive as $O(d^3)$ (cost of computing the Jacobian determinant for a general Jacobian matrix), if the associated Jacobian does not have a particular structure. However, normalizing flows are made very efficient by exploiting the fact that flexible bijections can be represented as a composition of simpler bijections, for which the individual Jacobian determinant can be computed efficiently.
>
> This is not the case for injective flows (where the dimensionality changes) because the resulting Jacobian determinant cannot be factored out in the product of the individual Jacobians. Therefore, the Jacobian determinant still requires $O(d^3)$ to be computed and current work focuses on approximations of the Jacobian determinant or consider a very restricted set of transformations. Instead, we show that for star-like manifolds we can exactly compute the Jacobian determinant in $O(d^2)$ (hence with the same overall complexity of normalizing flows).
>
> > Do you require any constraints on $T_\theta$?
>
> $T_\theta$ does not represent a bottleneck in our model since it can be implemented efficiently with any choice of normalizing flow layers. The bottleneck would normally come from computing $T_{s \rightarrow c} \circ T_r$, which is the injective part, but we show we can compute the associated Jacobian determinant efficiently (and exactly) in $O(d^2)$.
>
> > How do the properties of the manifold influence the training process of the flow model?
>
> In our experience, the model behaves very well independently of the manifold. When a new manifold is defined, the only part that needs to be changed explicitly is the function $r(\theta)$, which parametrizes the manifold (see Eqs. (25) and (26) in the Appendix). Some manifolds with extreme curvature might be more hard to learn, but this is a problem that is independent of the architecture used and more linked to the manifold itself. For instance, the lp-(pseudo) norm ball becomes quite challenging for $p<0.1$ and is characterized by extreme curvature (and most of its mass gets concentrated to the origin). This is, however, independent of our model implementation and more manifold-specific.
>
> [1] Caterini et al, Rectangular flows for manifold learning NeurIPS, 2021
>
> [2] Flouris et al., Canonical normalizing flows for manifold learning, NeurIPS, 2023
>
> [3] Sorrenson et al., Lifting architectural constraints of injective flows, ICLR, 2024

---

> > ### Comment · Reviewer_xkKJ · 2024-11-22
> >
> > Thank you for your responses and editing the manuscript. Based on the revised manuscript, I decide to raise my score to 6.

---

### Official Review · Reviewer_HG9C · 2024-11-04

**Soundness:** 4
**Presentation:** 4
**Contribution:** 2
**Rating:** 6
**Confidence:** 4

**Summary:**

A method is presented for efficiently computing normalizing flows on star-shaped d-manifolds embedded in (d+1)-dimensional Euclidean space. Using a decomposition based on spherical coordinates, the Jacobian determinant can be analytically computed in O(d^2) time. Numerical examples are given to show that the method outperforms the (d+1)-dimensional default sampler.

**Strengths:**

1. Paper is well written and readable.
2. The method is clearly explained and understandable, and it improves over sampling in the ambient dimension.
3. Claims are supported with proof and numerical evidence.

**Weaknesses:**

1. The method only applies to star-shaped manifolds embedded in one dimension higher. The authors acknowledge the limitation that their proof cannot be extended to manifolds with higher co-dimension. This is the main concern with the method - since this method only applies to a very specific class of co-dimension 1 submanifolds, I wonder about its applicability to problems of general interest. Since the method applies to a very specific class of manifolds instead of a more general class, the benefits for this specific class need to be greater to warrant acceptance into ICLR, in my opinion
2. The numerical examples seem limited (as a result of 1). To justify this approach, more examples of star-shaped manifolds and real-world applications should be presented and/or the benefit of the method over normal normalizing flows should be significant.

**Questions:**

1. The asymptotic efficiency of the method is shown to be O(d^2), it could be interested to look at the rate in more detail. Specifically, I am curious about the constant factors in the rate as compared to the normal normalizing flow methods. This can be done either analytically or empirically with comparisons of runtimes.

---

> ### Author Response · Authors · 2024-11-19
>
> We thank the reviewer for suggesting further experiments to strengthen our approach, which we implemented, and for highlighting that we should discuss further possible applications of our method.
>
> ## Weaknesses:
> > The method only applies to star-shaped manifolds embedded in one dimension higher. The authors acknowledge the limitation that their proof cannot be extended to manifolds with higher co-dimension. This is the main concern with the method; since this method only applies to a very specific class of co-dimension 1 submanifolds, I wonder about its applicability to problems of general interest. Since the method applies to a very specific class of manifolds instead of a more general class, the benefits for this specific class need to be greater to warrant acceptance into ICLR, in my opinion
>
> Our proposed method is indeed specialized to star-like manifolds and this does sound restrictive with respect to other injective flows. However, other competing methods can generalize to more manifolds because they approximate the Jacobian determinant. For the purpose of variational inference problems, we showed empirically that the exact Jacobian determinant is crucial. Currently, injective flows that allow for an exact Jacobian determinant computation focus on rather trivial manifolds [1] or use very restrictive transformations [2]. In contrast, our method extends to a whole class of manifolds. In the general response we also list some of the many applications where star-like manifolds are useful for variational inference and statistical analysis. We will highlight two important ones: Lasso regression problems and modeling distributions on the probabilistic simplex as with mixing models, for example. These are entirely separate fields of interest where our method is applicable.
>
>
> > The numerical examples seem limited (as a result of 1). To justify this approach, more examples of star-shaped manifolds and real-world applications should be presented and/or the benefit of the method over normal normalizing flows should be significant.
>
> We thank the reviewer for highlighting the need for further evaluation and for listing more possible applications. As part of the rebuttal, we compared our method with state-of-the-art method approaches to approximate the Jacobian determinant. In short, we show that current approaches to approximate the Jacobian determinant fail already for simple densities on trivial manifolds. In the global response we also list further potential applications for densities on star-like manifolds. It is important to note that we cannot compare our method to standard normalizing flow architectures as they are not designed for problems with injective steps where the dimensionality changes.
>
> ## Questions:
> > The asymptotic efficiency of the method is shown to be O(d^2), it could be interesting to look at the rate in more detail. Specifically, I am curious about the constant factors in the rate as compared to the normal normalizing flow methods. This can be done either analytically or empirically with comparisons of runtimes.
>
> We thank the reviewer for the very useful suggestion. We included an experiment where we measure the runtime of our method and compare it to the only possible alternative in the literature, which is the explicit computation in Eq. (2). We showed empirically that our method provides a significant speedup, in line with what we claimed in Theorem 1.
>
> [1] Rezende et al., Normalizing Flows on Tori and Spheres, ICML, 2020
>
> [2] Ross et al, Tractable density estimation on learned manifolds with conformal embedding flows, NeurIPS 2021

---

> > ### Comment · Reviewer_HG9C · 2024-11-25
> >
> > Thank you for the response and the additional experiment. I have raised my score.

---

### Official Review · Reviewer_pwAr · 2024-11-04

**Soundness:** 3
**Presentation:** 3
**Contribution:** 2
**Rating:** 6
**Confidence:** 4

**Summary:**

The authors propose a normalizing flow technique for a specific class of manifolds in which the computation of the Jacobian determinant is both exact and efficient, while related works rely on approximations or predefined manifolds. This approach is particularly relevant for variational inference problems where the manifold is known, and the goal is to approximate a density on it. The effectiveness of the method is demonstrated through variational inference on well-designed problems that meet the manifold constraints.

**Strengths:**

- The general idea of focusing on this class of manifolds, which has desired properties, is both simple and interesting, and the technical aspects appear solid.
- Based on the experiments conducted, the proposed model supports the claims.
- The paper is generally well-written and well-structured (see Questions for some suggestions).

**Weaknesses:**

- Even if this class of manifolds is useful for density modeling in the considered scenarios, it becomes somewhat restrictive if the $r(\theta)$ is not trainable.
- Further experiments and/or comparisons with related works, such as in a maximum likelihood setting, could have been beneficial for demonstrating both the effectiveness and efficiency of the approach.

**Questions:**

Q1. I believe Fig. 2 does not significantly aid in understanding the proposed approach, and a more intuitive figure could better illustrate the concept of the injective flow graphically. Additionally, moving the proof sketch to the appendix could free up space for a more intuitive figure(s), details on the mappings considered (e.g., as in Appendix A.5), and discussions on variational inference, maximum likelihood, and additional experiments.

Q2. The proposed approach appears to rely on the circular bijective layers of Rezende et al. (2020); a brief description in the appendix would be helpful. How does this approach scale in higher dimensions, and is this critical to the proposed approach?

Q3. Regarding experiment 5.1, is it possible that the density effectively modeled through the spherical flows (circular bijective layers) used for the transformation $\mathcal{T}_{\theta}$?

Q4. It would be interesting to include some (synthetic 2D or 3D) experiments where the manifold is more complicated, such as variational inference problems with a posterior concentrated on a star-shaped manifold, where $r(\theta)$ is potentially a learnable function.

Q5. Minor: What is meant by the density will be expressed in Cartesian coordinates (line 160)? Does this simply mean that the push-forward samples $x$ are just points in the ambient space?

---

> ### Author Response · Authors · 2024-11-19
>
> We thank the reviewer for suggesting interesting experiments and for the insights on how to restructure some parts of the paper.
>
> ## Weaknesses
> > Even if this class of manifolds is useful for density modeling in the considered scenarios, it becomes somewhat restrictive if the $r(\theta)$ is not trainable.
>
> The radial parameterization $r(\theta)$ could be made learnable only in maximum likelihood settings, where training is supported by observations. However, as we argue in the general response, star-like manifolds (codimension 1) are much more interesting for variational inference, where the goal is to learn some density on a manifold given some (unnormalized) target. Having the target density defined also means that the manifold is known. Many applications of statistical and practical interest lie in this category. We sketched some in the general response.
>
> > Further experiments and/or comparisons with related works, such as in a maximum likelihood setting, could have been beneficial for demonstrating both the effectiveness and efficiency of the approach.
>
> We thank the reviewer for the suggestion of including further experiments to assess (i) effectiveness and (ii) efficiency of the approach. We performed two additional experiments to assess both in a variational inference setting, which is the focus of our work.
>
> (i) As for the effectiveness, we showed that we significantly outperform competitive methods that use an approximate Jacobian determinant already on trivial densities on simple manifolds.
>
> (ii) As for the efficiency, we showed that computing the Jacobian determinant explicitly (Eq. (2)) leads to more than cubic complexity, while our proposed approach results in a significant speedup with sub-quadratic runtime.
>
> ## Questions
> > Q1. [...] Additionally, moving the proof sketch to the appendix could free up space for a more intuitive figure(s), details on the mappings considered (e.g., as in Appendix A.5), and discussions on variational inference, maximum likelihood, and additional experiments.
>
> We thank the reviewer for outlining possible restructuring of our work. The reason why we have proof sketch in the main paper is that we believe the proof of Theorem 1 is a fundamental contribution of our work. We agree that the discussion concerning maximum likelihood vs. variational inference (on the lines of what was argued in the general response) deserves a stand-alone paragraph. We did so in the new Section 3.3.
>
> > Q2. The proposed approach appears to rely on the circular bijective layers of Rezende et al. (2020); a brief description in the appendix would be helpful. How does this approach scale in higher dimensions, and is this critical to the proposed approach?
>
> The only layer that is specific to our model is the one that implements the transformation from spherical angles to cartesian coordinates ($T_{s \rightarrow c} \circ T_r$ in Figure 2). Else, the transformations $T_z \circ T_\theta$ can be implemented with any arbitrary bijection as long as $T_\theta$ maps to spherical angles. In our implementation, we tried several bijective layers and noticed we had only slightly better performance with circular splines.
>
> We thank the reviewer for suggesting to include a description of the circular spline layers. We included a longer description in the Appendix.
>
> > Q3. Regarding experiment 5.1, is it possible that the density effectively modeled through the spherical flows (circular bijective layers) used for the transformation $T_\theta$?
>
> We might not have fully understood the question, but we are happy to provide more insight on layers and learning procedures for Experiments 5.1 (now 5.2). In our architecture, the learnable layers are $T_z$ and $T_\theta$, $T_{s \rightarrow c}$ is fixed and known, while $T_r$ is specified according to the manifold under consideration (a sphere in Figure 4a/b and a deformed sphere in 4c).
>
> > It would be interesting to include some (synthetic 2D or 3D) experiments where the manifold is more complicated, such as variational inference problems with a posterior concentrated on a star-shaped manifold, where $r(\theta)$ is potentially a learnable function.
>
> We thank the reviewer for suggesting further 3D experiments on non-trivial densities on more complicated manifolds. We included such an experiment where we learned a sinusoidal density on a highly deformed sphere (Figure 4c). Also in this case, the model could accurately reconstruct the ground truth density. For the reasons outlined above and in the general response, we focused on variational inference settings where $r(\theta)$ is inherently known.
>
> > Q5. Minor: What is meant by the density will be expressed in Cartesian coordinates (line 160)? Does this simply mean that the push-forward samples are just points in the ambient space?
>
> Yes, we thank the reviewer for improving the clarity of that sentence. We updated the text accordingly.

---

> > ### Author Response · Authors · 2024-12-02
> >
> > Reviewer pwAr,
> >
> > Thank you very much for taking the time to review our paper.
> >
> > We hope that our responses have addressed your concerns satisfactorily. As the discussion period is nearing its end, we kindly ask you to let us know if there are any remaining points that require further discussion. If our responses have resolved your concerns, we would be grateful if you could consider revisiting your score.
> >
> > Thank you for your time and consideration.
> >
> > Sincerely, --Authors

---

### Author Response · Authors · 2024-11-19
**Global response**

# Global Response

We would like to thank all reviewers for acknowledging that the proposed model is novel and interesting, that the technical aspects are solid, and that we show interesting applications. We especially thank reviewers for suggesting further experiments, which we performed and which further highlighted the significance of our work.

In this global response, we describe the three additional experiments and we address common questions among reviewers: further possible applications and why we do not perform maximum likelihood training.

We have updated the pdf and color-coded in blue the changes we made.

**EDIT**: following the discussion with **reviewer xKG1**, we have included even further experiments:
1. a comparison with the approximate Jacobian as a function of the number of samples in the trace estimator (Figure 3b)
2. a comparison of the quality of the samples according to two metrics (Figure 10a and 10b) and a runtime comparison (Figure 10c)

These changes are color-coded in red.

## Additional experiments
### 1. Runtime comparison
As **reviewer HG9C**, **reviewer xkKJ** and **reviewer xKG1** suggested, we compare the runtime required to compute the Jacobian determinant with the proposed approach in Eq. (4) with the explicit computation in Eq. (2).

Results, shown in Figure 3a, indicate that a straightforward implementation using our theorem has a significantly lower runtime performance and empirically confirmed the claimed quadratic complexity (Theorem 1).

### 2. Approximate Jacobian comparison
Current work on injective flows [1,2,3], including SOTA [3], relies on the Hutchinson trace estimator to approximate the Jacobian. In contrast, for star-like manifolds, we can compute it exactly (and efficiently). As **reviewer xKG1** suggested, we show that for variational inference using the exact Jacobian is beneficial, even for simple densities and manifolds.

We train the same model with (i) the exact Jacobian given by Theorem 1 and with (ii) the trace estimator in [3]. We chose a very simple distribution (uniform) on a simple manifold ($l_p$ pseudo norm ball with $p=0.5$). This way we could evaluate the log density exactly and generate samples from the true distribution. As shown in Figure 3b, already in such a simple setting our method is significantly more accurate than the approximate one, even when the number of samples in the trace estimator is increased.

### 3. Non-trivial manifold
As **reviewer pwAr** suggested, we included an additional 3D illustrative example for a non-trivial density on a non-trivial manifold. As shown in Figure 4c, also in this case the proposed method can accurately reconstruct the target density.

## Further possible applications
As opposed to other injective methods with exact Jacobian, we are not restricted to very trivial manifolds [4] or very trivial transformations [5]. Instead, our method applies to the whole class of star-like manifolds, which are very useful in various settings:
- statistics on hypersphere: directional statistics, astrophysics, medicine, biology, meteorology etc.
- oblate spheroid: to accurately model densities on Earth
- probabilistic simplex: Bayesian tracer mixing models, probabilistic portoflio analysis, probabilistic treatment of archetype models etc.
- l_p (pseudo) norm balls: objective Baysian treatment of Lasso problems. Can also be extended to the generalized lasso, which includes as a special case fused lasso and standard lasso

For all these application, our model improves significantly the runtime (compared to other exact methods) and significantly improves the density approximation (compared to approximate ones)

## Why no maximum likelihood training
In this paper we focus on variational inference settings (without observations) and not on the maximum likelihood setting (with observations) for two reasons:
1. the present method works for star-like manifolds (co-dimension 1), which are more useful in variational inference settings. In maximum likelihood applications, the underlying manifolds are often assumed to be much lower dimensional. We therefore focus on the variational inference as it is more reflected in applications.
2. while maximum likelihood settings approximate Jacobian determinants work well in practice, in variational inference the exact Jacobian determinant is crucial in order to learn the target distribution. We showed empirically that approximating the Jacobian determinant worsens the performance already for trivial densities on simple manifolds (Figure 3b).

[1] Caterini et al, Rectangular flows for manifold learning NeurIPS, 2021

[2] Flouris et al., Canonical normalizing flows for manifold learning, NeurIPS, 2023

[3] Sorrenson et al., Lifting architectural constraints of injective flows, ICLR, 2024

[4] Rezende et al., Normalizing Flows on Tori and Spheres, ICML, 2020

[5] Ross et al, Tractable density estimation on learned manifolds with conformal embedding flows, NeurIPS 2021

---

### Author Response · Authors · 2024-12-03
**Short summary of the rebuttal**

We would like to express once again our gratitude to the reviewers for their insightful feedback, which we believe has strengthened our results and the overall quality of the paper. We summarize below the novelty and relevance aspects of our approach and how we addressed reviewers' concerns by running additional experiments.

### Summary of proposed approach
1. **Theoretical contribution**. In Theorem 1 we show that for star-like manifolds we can compute the Jacobian determinant exactly and efficiently. This was not possible before our work. We improve the computational complexity from $O(d^3)$ to $O(d^2)$
2. **Empirical validation and motivation**:
We empirically test the computational speedup (**Figure 3a**) and show that the exact Jacobian determinant is crucial in variational inference settings (the focus of our work), already in simple settings (**Figure 3b**)
3. **(Novel) Applications**:
We argue that star-like manifolds are useful in many applications of interest in statistical analysis and variational inference settings. We then explicitly showcase the proposed approach in novel applications: **Objective Bayesian Lasso** (Figure 5 and 6) and **Bayesian mixing models** (Figure 7)

### Addressing reviewers' concerns
We ran **additional experiments** to quantitatively address concerns regarding the following aspects:
1. **Runtime comparison**: As suggested by reviewers **HG9C**, **xkKJ** and **xKG1**, we empirically tested Theorem 1 and showed experimentally that we indeed obtain the expected computational speedup from $O(d^3)$ to $O(d^2)$ (**Figure 3a**).
2. **Comparison with approximate Jacobian**: As suggested by reviewer **xKG1**, we show that in variational inference settings (the focus of our work) the exact Jacobian determinant is crucial, even for simple densities and manifolds (**Figure 3b**)
3. **Non-trivial manifold example**: As reviewer **pwAr** suggested, we included an additional 3D illustrative example for a non-trivial density on a non-trivial manifold (**Figure 4c**).

We addressed a number of further minor concerns. Details can be found in the global response and within each individual review.

---

### Meta-Review · Area_Chair_VyXf · 2024-12-23

**Metareview:**

The paper proposes injective flows for star-like manifolds of dimension $d-1$ in Euclidean space $\mathbb{R}^d$ and show that for such manifolds the Jacobian determinant can be computed exactly and efficiently, with the same computational cost as Normalizing Flows (NFs).
The proposed model is supported by several numerical experiments.

Reviewers generally agree that the paper is technically solid, the idea presented interesting, and the presentation is clear and well-structured. I agree that the proposed framework, where Jacobian can be computed exactly, is of sufficient interest to recommend acceptance.

**Additional Comments On Reviewer Discussion:**

There were extensive discussions between the authors and reviewers, including the following points

- Reviewer HG9C was concerned about the scope of applicability: the current method applies specifically to star-shaped manifolds of co-dimension 1. In their response, the authors pointed out that other competing methods work on more general manifolds but need to approximate the Jacobian determinant. They demonstrated experimentally that the exact Jacobian determinant is crucial in variational inference settings. They also pointed out examples of applications for their framework in variational inference and statistical analysis, including Lasso regression problems and modeling distributions on the probabilistic simplex as with mixing models.

- All reviewers suggested further numerical experiments. In their response, the authors provided some additional experiments, including demonstration of runtime speed up and comparison with approximate Jacobian. They also noted that the current method to cannot be compared with standard normalizing flow architectures as they are not designed for problems with injective steps where the dimensionality changes.

After the discussion period, all of the reviewers were satisfied by the various clarifications and all raised their scores to 6, to which I concur.

---

### Decision · Program_Chairs · 2025-01-22

Accept (Poster)